# Genetic relationships of *Aspergillus fumigatus* in hospital settings during COVID-19

Raeseok Lee,[1,2] Won-Bok Kim,[2,3] Sung-Yeon Cho,[1,2] Dukhee Nho,[1,2] Chulmin Park,[2] Hye-Sun Chun,[2] Jun-Pyo Myong,[4] Dong-Gun Lee[1,2]

**ABSTRACT**   The transmission pathways and risks of COVID-19-associated pulmonary aspergillosis (CAPA) remain unclear. This study investigated the genetic relationships of *Aspergillus fumigatus* isolates from patients with and without COVID-19 and environmental air samples to suggest possible transmission patterns. We conducted a prospective study from March 2020 to December 2022, collecting clinical and environmental isolates from a tertiary hospital. Isolates from patients with and without COVID-19 were compared with those from air samples at four hospital locations. The genetic analysis included internal transcribed spacer and *β-tubulin A* sequencing, with azole resistance assessed via *cyp51A* gene analysis. Multiple locus variable-number tandem repeat analysis was performed to elucidate genetic relationships. A total of 155 isolates (19 from COVID-19 patients, 104 from non-COVID-19 patients, and 32 from environmental samples) were identified and genotyped, revealing 131 sequence types (Simpson Diversity Index 0.9972). Four CAPA clinical strains genetically related to environmental strains were isolated from the COVID-19 intensive care unit (ICU), while two CAPA clinical strains sharing multiple locus variable-number tandem repeat sequence types and azole-resistant mutations were isolated in the same COVID-19 ICU 4 months apart. All but one of these strains were isolated from patients requiring mechanical ventilation. The observed genetic similarities between strains from critically ill patients with COVID-19 and those from the environment, as well as within the same ICU, raise the possibility of nosocomial acquisition via contaminated air or environmental sources. These findings highlight the risks of CAPA associated with negative pressure rooms and the need for enhanced environmental infection control measures.

**IMPORTANCE**   This study reveals genetic links between *Aspergillus fumigatus* in patients with COVID-19 and environmental sources, suggesting nosocomial transmission and urging a reevaluation of universal negative pressure isolation practices in hospitals, especially for critically ill patients.

**KEYWORDS**   *Aspergillus*, pulmonary aspergillosis, COVID-19, molecular epidemiology, healthcare-associated pneumonia

Invasive pulmonary aspergillosis (IPA) is a life-threatening opportunistic fungal infection caused by the inhalation of *Aspergillus* spores (1, 2). IPA predominantly affects severely immunocompromised individuals, including those with hematological malignancies (3, 4). However, the incidence among critically ill patients in intensive care unit (ICU), who were previously not considered high-risk, is increasing (5). This increase is particularly concerning among patients with severe respiratory infections, such as COV-19 and influenza, presenting a novel and significant public health challenge (6, 7). Reports indicate that COVID-19-associated pulmonary aspergillosis (CAPA) affects 3%–33% of critically ill patients with COVID-19 (6, 8).

**Peer Reviewer** Eveline Snelders, Wageningen University & Research, Wageningen, the Netherlands

Address correspondence to Dong-Gun Lee, symonlee@catholic.ac.kr.

Raeseok Lee and Won-Bok Kim contributed equally to this article. Author order was determined in order of increasing seniority.

The authors declare no conflict of interest.

See the funding table on p. 10.

*Aspergillus* spores are ubiquitous in indoor and outdoor environments, including hospitals, and have been linked to a potential increase in hospital-acquired IPA among immunocompromised patients, particularly from environmental sources like construction sites (9, 10). Concerns have been raised regarding the risk of CAPA in critically ill patients with COVID-19 treated in negative pressure rooms, which may enhance the likelihood of inhaling *Aspergillus* spores (11, 12). Despite ongoing debates and clinical urgency, significant gaps remain in our understanding of the epidemiological and molecular relatedness of *Aspergillus* in these settings, particularly regarding whether infections are acquired from the community or develop nosocomially (13, 14).

Here, we used multiple locus variable-number tandem repeat analysis (MLVA) to examine the genetic diversity of clinical and environmental *Aspergillus fumigatus* isolates. Our aim was to explore potential epidemiological and molecular relationships between *A. fumigatus* isolates from patients with and without COVID-19, as well as from environmental air samples collected over a 3-year period during the COVID-19 pandemic. Additionally, we assessed the possible implications of using negative-pressure environments during the pandemic.

## MATERIALS AND METHODS

### Study design and hospital setting

We conducted a prospective study at Seoul St. Mary's Hospital, a 1,350-bed tertiary university-affiliated hospital, from March 2020 to December 2022, coinciding with the peak of the COVID-19 pandemic. The hospital functioned as a primary referral center for critically ill patients with COVID-19, both from the local community and other medical institutions. In response to the pandemic, an ICU for patients with critical COVID-19 (COVID-ICU) was established, comprising 14 beds across six units (Fig. 1). This COVID-ICU featured high-efficiency particulate air (HEPA)-filtered air circulation with an independent ventilating system that was maintained under negative pressure. The study was approved by the ethical review board of Seoul St. Mary's Hospital (no. KC22SISI0481) and was conducted in accordance with the Declaration of Helsinki, 2013.

### Case definition and data collection

We prospectively gathered *A. fumigatus* isolates from COVID-ICU patients and from patients without COVID-19 (Fig. 2). CAPA diagnosis followed the 2020 consensus definition of the European Confederation of Medical Mycology and the International Society for Human and Animal Mycology (7). For patients without COVID-19, we used criteria from the European Organization for Research and Treatment of Cancer/Invasive Fungal Infections Cooperative Group and the National Institute of Allergy and Infectious Diseases Mycoses Study Group, along with *Asp*ICU criteria (15–17). Diagnoses of CAPA or IPA were confirmed by consensus between two independent infectious disease experts. Clinical strains were classified as "proven," "probable/putative," or "possible" pathogens, while "none" indicated colonization (15–17). No antifungal prophylaxis was used for patients with COVID-19 during the study, except for posaconazole prophylaxis in patients with acute myeloid leukemia or myelodysplastic syndrome undergoing remission induction intensive chemotherapy or graft-vs-host disease post-HSCT (18).

### Environmental air sampling and culture

We conducted biweekly air sampling indoors and outdoors at the hospital in central metropolitan Seoul, which is adjacent to two parks and a riverside (9). This was performed between March 2021 and February 2022 at four distinct locations to account for seasonal variations: at two sites inside COVID-ICU patient rooms, one near the elevator on the same floor, and one outside the hospital's main building (Fig. 1). At each site, air sampling was performed three times at 20 min intervals. Further details are provided in the Supplementary methods.

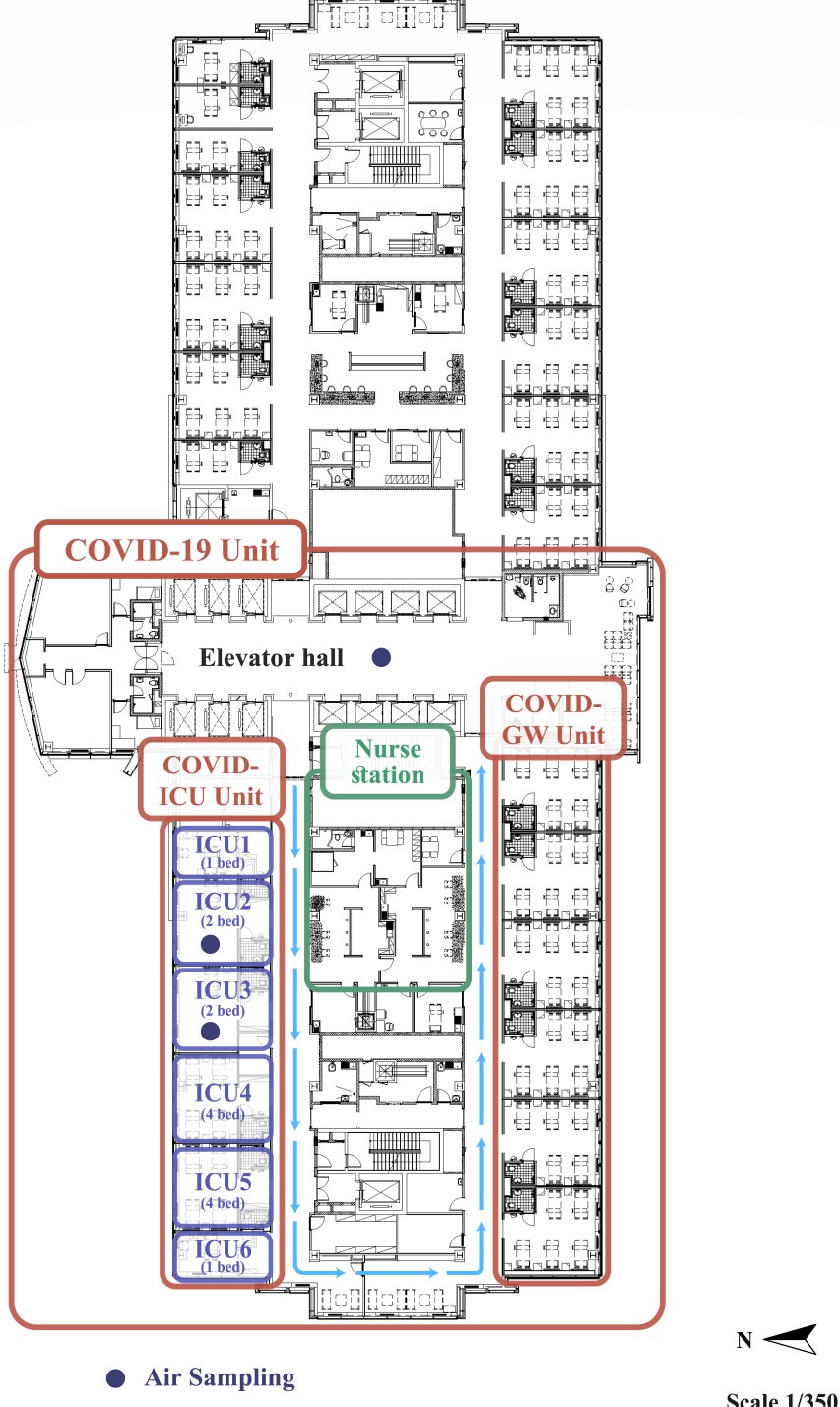

**FIG 1** Indoor air sampling locations and map of the COVID-19 ICU. COVID-GW, COVID-19 general ward.

## Identification and susceptibility testing

After microscopic examination, *A. fumigatus* isolates were obtained from the environmental air samples, and single-colony pure cultures of selected clinical strains were prepared. DNA was extracted from the conidia of all isolates using the procedure detailed in a previous study (19). Furthermore, the phenotypic response to azoles was assessed through MIC analysis using the *A. fumigatus* isolates. Additionally, we analyzed

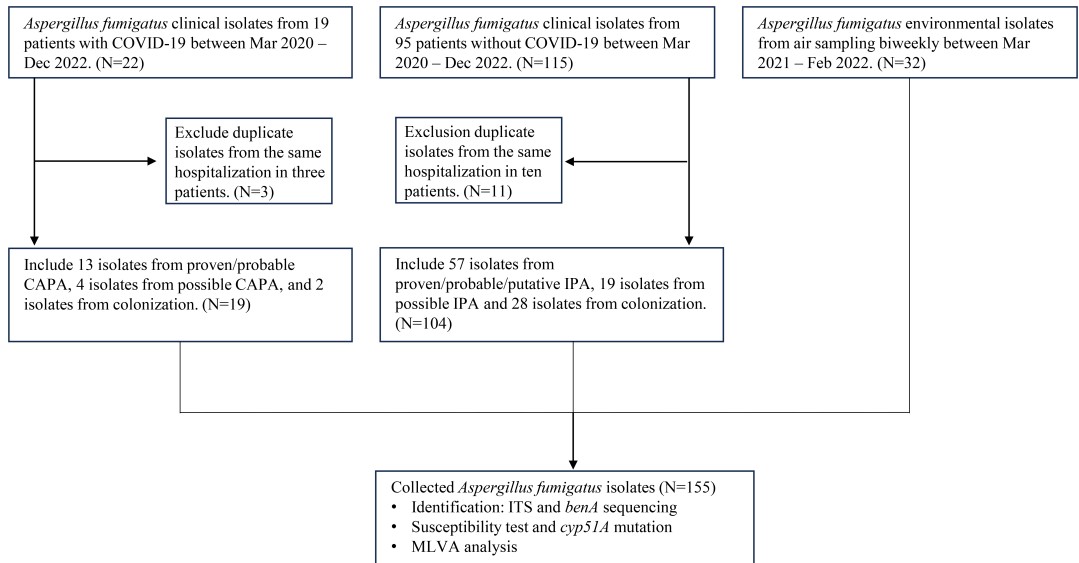

**FIG 2** Study flow and isolate inclusion. ITS, internal transcribed spacer.

the *cyp51A* gene, which is associated with azole resistance, to assess its resistance profile. Detailed experimental methods are described in the Supplementary methods and Table S1.

## Molecular typing

Genotyping of the selected *A. fumigatus* strains was conducted using the MLVA method to analyze 10 variable-number tandem repeat (VNTR) polymorphisms identified in prior studies (10, 20–23). The VNTR markers, tagged using fluorescent dye-labeled primers (detailed in Table S2), were amplified using 1–5 ng of template DNA and the SuperPlex Premix kit (Takara Korea Biomedical Inc., Seoul, Korea), following the manufacturer's PCR protocol. Multiplex PCR was performed three times, and the resulting products were analyzed for fragment sizes of each VNTR locus at MACROGEN Co. (Seoul, Korea). The number of tandem repeat (TR) sequences at each VNTR locus was calculated by dividing the difference in base pair length (excluding the TR) from the analyzed fragment size by the size of the TR, and the results from all three fragment analyses were consistent. The sequence type (ST) was determined by aligning the results from the 10 markers.

The minimum spanning tree (MST) was generated using the PHYLOVIZ V2.0 (Lisboa, Portugal) to examine relationships among isolates, using Euclidian and goeBURST distances for MST and cluster analysis, respectively (24). The discriminatory power was evaluated using the Simpson Diversity Index (SDI). Dendrograms were generated using the unweighted pair-group method with arithmetic means.

## RESULTS

### Selection of clinical and environmental isolates

*Aspergillus* species were prospectively collected at our hospital between January 2020 and December 2022. Nineteen *A. fumigatus* strains were isolated from 19 patients with COVID-19 and 104 strains from 95 patients without COVID-19 (Fig. 2). The clinical characteristics were comparable between the two patient groups. Most of the isolates (94%) were obtained from respiratory samples, and 75.4% came from patients diagnosed with at least "possible" invasive aspergillosis (Table 1).

We analyzed 251 conidia from environmental samples and identified 32 *A. fumigatus* isolates. The detection frequency of *A. fumigatus* was as follows: 1 from 55 isolates collected in patient rooms, 2 from 52 collected in elevator hallways, and 29 from 144 collected in outdoor area, with a minimum count of 1 colony-forming unit (CFU)/m$^3$.

The analysis of all *A. fumigatus* isolates obtained, regardless of environmental or clinical sources, revealed a total of 155 isolates: 36 in spring, 42 in summer, 48 in fall, and 29 in winter. Therefore, it was confirmed that the occurrence of *A. fumigatus* shows no significant variability due to seasonal changes. Variation between measurements was noted, especially in the outdoor areas according to humidity (mean fungal conidial count: 115.9 CFU, range: 2–400 CFU). Areas equipped with HEPA filters, such as patient rooms, showed minimal fluctuations (mean: 1.8 CFU, range: 0–7.5 CFU; Fig. S1).

## Molecular identification and azole resistance

Phylogenetic analysis using internal transcribed spacer and *β-tubulin A* gene sequencing was conducted on 155 isolates. These tests confirmed that all the samples were *A. fumigatus*, although they did not allow for molecular differentiation between isolates (Fig. S2). Minimum inhibitory concentration testing indicated that most strains were susceptible to azoles. However, azole resistance was observed in seven clinical isolates, representing an incidence rate of approximately 5% across both COVID-19 and non-COVID-19 patient samples. In contrast, no azole resistance or *cyp51A* gene mutations were found in the environmental samples (Table 2).

**TABLE 1** Baseline characteristics of the study participants[a,b]

| Variables | Patients without COVID-19 (*N* = 95) | Patients with COVID-19 (*N* = 19) | Total (*N* = 114) | *P* value |
|---|---|---|---|---|
| Age, median with IQR | 67.0 (60.0;76.5) | 65.0 (60.0;74.0) | 67.0 (60.0;76.0) | 0.982 |
| Sex (male) | 56 (58.9) | 15 (78.9) | 71 (62.3) | 0.167 |
| Hematological malignancies | 28 (29.5) | 5 (26.3) | 33 (28.9) | 1.000 |
| Immunosuppression | 57 (60.0) | 15 (78.9) | 72 (63.2) | 0.193 |
| Oxygen supplement | | | | 0.176 |
| None or nasal cannula | 44 (46.3) | 5 (26.3) | 49 (43.0) | |
| HFNC or MV | 51 (53.7) | 14 (73.7) | 65 (57.0) | |
| Specimen (respiratory) | 89 (93.7) | 18 (94.7) | 107 (93.9) | 1.000 |
| Location (ICU) | 31 (32.6) | 12 (63.2) | 43 (37.7) | 0.025 |
| Classification (hospital acquired) | 79 (83.2) | 16 (84.2) | 95 (83.3) | 1.000 |
| IPA or CAPA category | | | | 0.245 |
| None (colonization) | 26 (27.4) | 2 (10.5) | 28 (24.6) | |
| Possible | 12 (12.6) | 4 (21.1) | 16 (14.0) | |
| Proven/probable/putative | 57 (60.0) | 13 (68.4) | 70 (61.4) | |
| GM_BAL | | | | 0.071 |
| Not available | 66 (69.5) | 18 (94.7) | 84 (73.7) | |
| Negative | 7 (7.4) | 0 (0.0) | 7 (6.1) | |
| Positive | 22 (23.2) | 1 (5.3) | 23 (20.2) | |
| GM_serum | | | | 0.003 |
| Not available | 22 (23.2) | 1 (5.3) | 23 (20.2) | |
| Negative | 42 (44.2) | 4 (21.1) | 46 (40.4) | |
| Positive | 31 (32.6) | 14 (73.7) | 45 (39.5) | |
| Treatment | | | | 0.115 |
| None | 41 (43.2) | 7 (36.8) | 48 (42.1) | |
| Mold active azole | 38 (40.0) | 11 (57.9) | 49 (43.0) | |
| Amphotericin B | 15 (15.8) | 0 (0.0) | 15 (13.2) | |
| Echinocandins | 1 (1.1) | 1 (5.3) | 2 (1.8) | |
| Azole resistance | 4 (4.2) | 1 (5.3) | 5 (4.4) | 1.000 |
| Overall mortality | 40 (42.1) | 10 (52.6) | 50 (43.9) | 0.555 |

[a]Values are presented as number (%), median (interquartile range).
[b]BAL, bronchoalveolar lavage; CAPA, COVID-19-associated pulmonary aspergillosis; GM, galactomannan; HFNC, high-flow nasal cannula; ICU, intensive care unit; IPA, invasive pulmonary aspergillosis; IQR, interquartile range; ITS, internal transcribed spacer; MLVA, multiple locus variable-number tandem repeat; MV, mechanical ventilation.

## Genetic diversity between isolates

MLVA of 155 isolates revealed 131 unique STs, with 24 appearing more than once. The SDI was 0.9972, demonstrating high genetic diversity with a 95% CI of 0.9955–0.998. We have summarized all the results used in the MLVA and included them as a Supplementary file.

MST revealed three major groups of isolates. Isolates from patients with COVID-19 (marked with a red circle) were distributed across all groups; however, two of the isolates (F442 and F573) were closely related (Fig. 3 and 4). Similar sequence types were observed between an isolate from a patient with COVID-19 (F557) and one from a patient without COVID-19 (F492). Several isolates from patients without COVID-19 and environmental samples also displayed genetic relationships (within three VNTR marker differences) with those isolated from patients with COVID-19 (Fig. 3 and 4).

In Fig. 3, arrows and annotations have been added to indicate azole-resistant isolates. Isolates without resistance labels are azole susceptible. These resistant isolates were predominantly found within the same group, but one genetically distinct azole-resistant isolate (F501) without TR mutations was in a different group.

## Epidemiological and molecular relationships among isolates

We performed a comprehensive analysis of the hierarchical clustering dendrogram, focusing on groups containing isolates from patients with COVID-19 with a cut-off of 1.5, corresponding to three VNTR marker differences. The related isolates are shown in Fig. 4.

In clusters 1, 4, and 9, the strains from patients with CAPA were closely matched to those in the outdoor air samples. In clusters 4 and 9, genetically related strains from environmental samples were isolated from patients with COVID-19 within a 1-month interval. In cluster 3, a total of 18 instances of genetically identical or similar strains were identified from clinical and environmental samples collected in close temporal and spatial proximity, based on VNTR analysis and point mutation results within the *cyp51*A gene (Fig. 4). These strains were initially detected at the entrance of the COVID-19 unit and subsequently isolated from patients diagnosed with CAPA. All strains associated with CAPA in these clusters were obtained from patients requiring mechanical ventilation (MV).

Clusters 12 and 13 revealed genetically related strains among different patients within the same COVID-19 unit, indicating potential intra-unit environmental transmission. In particular, cluster 13 revealed CAPA cases involving genetically similar strains (F573 and F442), which displayed a single VNTR difference and shared the same point mutation within the *cyp51*A gene. Similar to clusters 1, 3, 4, and 9, all strains, except F664 in cluster 13, were from patients with CAPA who were also on MV.

## DISCUSSION

Our study revealed that *A. fumigatus* strains isolated from patients with COVID-19 exhibited genetic relatedness to strains from patients without COVID-19 and environmental sources, indicating the possibility of widespread nosocomial infections via contaminated air. Furthermore, the detection of genetically related strains within the COVID-ICU raises concerns about the potential for cross contamination. Notably, the majority of suspected nosocomial infections, whether from contaminated air or direct cross contamination, occurred in patients requiring MV. These findings suggest that the widespread use of negative pressure rooms during the COVID-19 pandemic may not have been the optimal approach for protecting this vulnerable group from CAPA. It is essential to reconsider isolation practices in negative-pressure environments to enhance safety and infection control measures for high-risk patients.

While negative air-pressure isolation rooms are routinely recommended in the management of COVID-19, there are concerns that it may contribute to the occurrence of CAPA (11, 12). In addition, the appropriateness of the use of negative pressured ICU,

**TABLE 2** Azole susceptibility and mutation profiles of the clinical and environmental *A. fumigatus* isolates[a,b]

| Source and azole susceptibility | TR | *cyp51A* mutation (number) | MIC (µg/mL) | | | Category | Outcome | |
|---|---|---|---|---|---|---|---|---|
| | | | ITC | VRC | PSC | | Treatment | Outcome |
| Clinical, COVID-19 (n = 19) | | | | | | | | |
| Susceptible (n = 18) | (–)[c] | None (15) | 0.125–0.5 | 0.125–1 | 0.06–1 | | | |
| | (–) | F46Y, M172V, and E427K | 0.25 | 0.5 | 0.06 | Possible | CAF | Death |
| | (–) | F46Y, M172V, and E427K | 0.5 | 0.5 | 0.125 | Probable | None | Death |
| | (–) | Q312H | 0.5 | 0.125 | 0.125 | Possible | None | Survive |
| Resistant (n = 1) | TR46 | Y121F, P216S, T289A, S363P, I364V, and G448S | **32** | **64≤** | **32** | Probable | VRC | Death |
| Clinical, Non-COVID-19 (n = 104) | | | | | | | | |
| Susceptible (n = 98) | (–) | None (79) | 0.06–1 | 0.06–1 | 0.06–0.5 | | | |
| | (–) | N248K | 0.25 | 0.25 | 0.125 | None | None | Survive |
| | (–) | N248K | 0.25 | 0.25 | 0.125 | Probable | VRC | Death |
| | (–) | F46Y, M172V,N248T, D255E, and E427K | 0.5 | 1 | 0.25 | Possible | None | Survive |
| | (–) | N248K | 0.25 | 0.25 | 0.125 | None | None | Survive |
| | (–) | N248K | 0.5 | 0.25 | 0.5 | None | None | Survive |
| | (–) | F46Y, M172V, N248T, D255E, and E427K | 0.25 | 0.06 | 0.25 | Possible | None | Survive |
| | (–) | F46Y, M172V, and E427K | 0.25 | 0.06 | 0.25 | None | None | Survive |
| | (–) | N248K | 0.125 | 0.06 | 0.25 | None | None | Survive |
| | (–) | N248K | 0.25 | 0.25 | 0.125 | Possible | None | Survive |
| | (–) | N248K | 0.125 | 0.125 | 0.125 | Probable | VRC | Death |
| | (–) | M39I | 0.5 | 0.25 | 0.125 | Possible | None | Death |
| | (–) | M39I | 0.25 | 0.25 | 0.125 | Probable | VRC | Survive |
| | (–) | F46Y, M172V, N248K, D255E, and E427K | 0.25 | 0.5 | 0.06 | None | None | Survive |
| | (–) | A9T | 0.25 | 0.25 | 0.06 | None | None | Survive |
| | (–) | N248K | 0.25 | 0.25 | 0.06 | Probable | VRC | Death |
| | (–) | A9T | 0.5 | 0.5 | 0.125 | Probable | VRC | Survive |
| | (–) | S335H | 0.5 | 0.25 | 0.125 | None | None | Survive |
| | (–) | N248K | 0.5 | 0.25 | 0.125 | Probable | LAB | Death |
| | (–) | M220T | | | | Possible | None | Survive |
| Resistant (n = 6) | (–) | None (1) | **2** | 0.5 | 0.25 | Possible | None | Survive |
| | TR34 | L98H | **4** | **4** | 1 | Possible | VRC | Death |
| | TR34 | L98H, S297T, and F495I | **64≤** | 1 | **2** | Probable | LAB | Death |
| | TR34 | L98H, S297T, and F495I | **64** | 1 | 1 | Probable | VRC | Death |
| | TR34 | L98H, S297T, and F495I | **16** | **4** | **2** | None | 0 | Survive |
| | TR46 | Y121F and T289A | 1 | **64** | 0.5 | Probable | VRC | Death |
| Environment (n = 32) | | | | | | | | |
| Susceptible (n = 32) | (–) | None (32) | 0.06–0.5 | 0.06–1 | 0.06–0.25 | NA | NA | NA |

[a]Bold values indicate MIC values considered to represent azole resistance.
[b]CAF, caspofungin; ITC, itraconazole; LAB: liposomal amphotericin B; MIC, minimal inhibitory concentration; NA, not available; PSC, posaconazole; TR, tandem repeat; VRC, voriconazole.
[c]"(-)" means that no TR was found.

particularly for critically ill or immunocompromised patients with COVID-19 who are more susceptible to IPA, remains controversial (12, 25). Unlike typical hospital-acquired IPA, which is often linked to air contamination from construction activities, the specific transmission pathways of *Aspergillus* during the COVID-19 pandemic are not well defined (10, 13, 26).

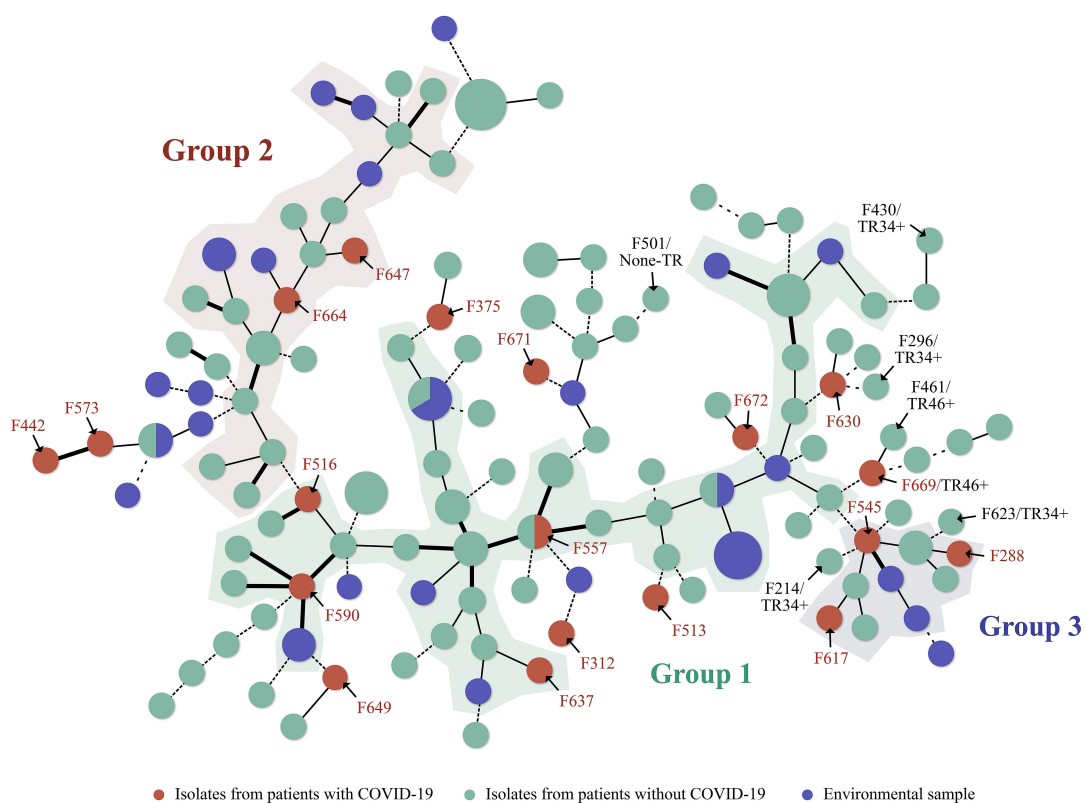

**FIG 3** A tree displaying 131 STs (circles) derived from 155 isolates. Circles are color coded by source: red represents isolates from patients with COVID-19, light green represents isolates from patients without COVID-19, and blue represents environmental samples. Circle size indicates the number of isolates sharing the same ST. Lines connecting the circles depict genetic distances, with bold lines representing a one-marker difference, solid lines a two-marker difference, and dotted lines a three-marker difference. Azole-resistant strains are marked by TR mutations, along with their corresponding strain numbers.

Throughout the COVID-19 pandemic, our molecular and spatiotemporal analysis of *A. fumigatus* demonstrated that strains from patients with CAPA were genetically identical or closely related to those isolated from patients without COVID-19 and environmental air samples. This supports the likelihood of airborne nosocomial transmission, especially since specific STs found temporally and spatially related in patients with non-COVID-19, and hospital environments were also identified in patients with CAPA. Predominantly, these cases involved critically ill patients with COVID-19 requiring MV. However, there was no direct correlation between high environmental *Aspergillus* burdens and the isolation of *A. fumigatus* from patients with COVID-19, indicating that individual host factors may be more critical than environmental fungal loads in the acquisition of hospital-acquired infections.

Research indicates that airborne fungal burdens are comparable in rooms with neutral and positive air pressures (27). Based on our results and these findings, we recommend that high-risk groups, such as critically ill or immunocompromised patients, needed to be treated in neutral air pressure isolation rooms equipped with HEPA filters. This approach appears to provide a more effective balance between infection control and patient safety, minimizing the risk of nosocomial infections while ensuring a safer environment for both patients and healthcare staff (14, 27).

During the COVID-19 pandemic, the rapid increase in patient admissions and constrained hospital capacities raised significant concerns about bacterial and fungal cross contamination within COVID-19 units (28–30). While previous studies have not conclusively proven hospital-acquired infections due to cross contamination, our research identified cases of CAPA involving genetically related strains within the same COVID-19 unit (31). Notably, in the COVID-ICU, where air was exchanged six times per

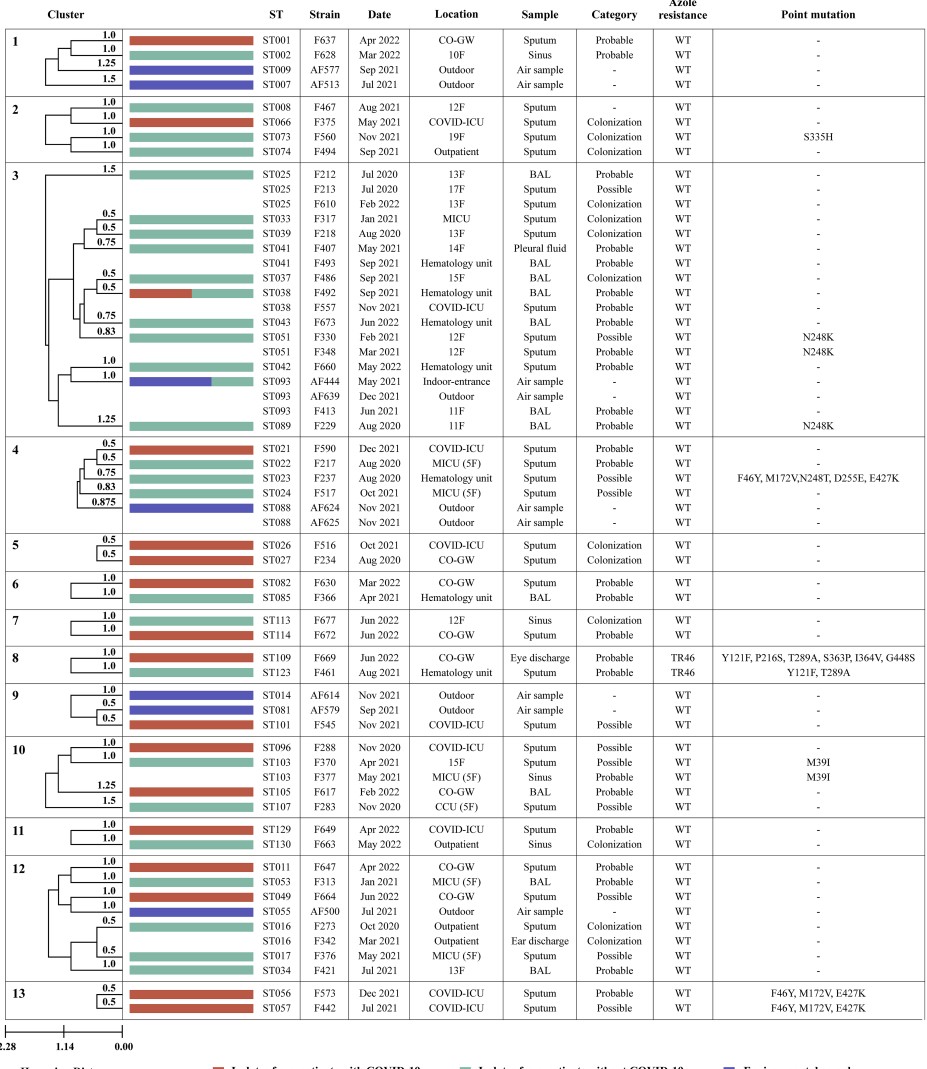

**FIG 4** Clustering dendrogram of isolates from patients with COVID-19. Constructed using the PHYLOVIZ 2.0 tool based on the unweighted pair group method with arithmetic mean analysis, this dendrogram displays clusters of isolates from patients with COVID-19 with other clinical or environmental isolates that met the cutoff value of 1.5. Notably, in clusters 1, 3, 4, and 9, strains from patients with CAPA closely matched genetically and spatiotemporally those found in outdoor air samples. Clusters 12 and 13 feature genetically related strains isolated from different patients within the same COVID-19 unit, highlighting potential environmental transmission. BAL, bronchoalveolar lavage; CO-GW, COVID-19 general ward; F, floor; MICU, medical ICU; WT, wild type.

hour, strains matching in VNTR analysis and resistance mutations were detected in the same room four months apart, suggesting the possibility of hospital-acquired infections linked to environmental contamination. Importantly, all affected patients except one were critically ill and required MV, consistent with other cases of hospital-acquired infections attributed to contaminated air.

In the management of severe respiratory viral infections, the focus on protective environments often centers around ventilation. However, our study shows that the risk of cross contamination due to environmental factors is considerable and should not be overlooked. Consequently, for patients at high risk of IPA, it is crucial to adopt comprehensive infection control strategies that extend beyond the optimization of ventilation systems (32). Implementing these multidimensional measures, including ventilation

system and environmental cleaning, is crucial not only for enhancing patient safety but also for significantly reducing the incidence of hospital-acquired infections.

This study had some limitations. First, the collection of adequate *Aspergillus* strains from patients with COVID-19 was hindered, potentially due to the low-culture sensitivity of *Aspergillus* species from clinical specimens and reluctance to perform invasive procedures like bronchoscopy because of the risk of viral transmission (6, 33). Second, our study was also limited by a low number of environmental isolates, which are essential for robust genetic and spatiotemporal analyses (10). Although our air sampling was extensive, lasting >1 year, to account for variations due to temperature and humidity, it did not cover all COVID-19 units or encompass many indoor and outdoor locations (34). Additionally, our efforts were focused within the COVID-ICU, where stringent infection controls, such as rapid ventilation systems with HEPA filters and enhanced environmental cleaning, may have reduced the culture sensitivity of *Aspergillus* from air sampling, further narrowing the scope of our analysis (14).

In conclusion, this study elucidates the genetic relationships between *A. fumigatus* strains isolated from patients with COVID-19, patients without COVID-19, and environmental sources. Our findings emphasize the substantial risk of nosocomial transmission within negative pressure environments, particularly for critically ill patients. Instead of universally applying negative pressure rooms, adopting tailored strategies is crucial to effectively mitigate the risk of CAPA.

## ACKNOWLEDGMENTS

We acknowledge the Department of Occupational and Environmental Medicine for their consultation as well as statistical and technical support for this study.

This work was supported by the Basic Science Research Program through the National Research Foundation of Korea (NRF) funded by the Ministry of Education (NRF-2022R1I1A01070887).

R.L., W.B.K., D.G.L., and J.P.M. conceptualized and R.L., C.P., S.Y.C., D.G.L., and J.P.M. coordinated this study. R.L., W.B.K., S.Y.C., H.S.C., and D.N. performed environmental sampling and data analysis. R.L., W.B.K., S.Y.C., D.N., C.P., and D.G.L. interpreted the data. R.L. and W.B.K. drafted the manuscript, and all authors have fully reviewed the manuscript. All authors agree with all contents and conclusions of this manuscript.

## AUTHOR AFFILIATIONS

[1]Division of Infectious Diseases, Department of Internal Medicine, College of Medicine, The Catholic University of Korea, Seoul, Republic of Korea
[2]Vaccine Bio Research Institute, College of Medicine, The Catholic University of Korea, Seoul, Republic of Korea
[3]Department of Biomedicine and Health Sciences, College of Medicine, The Catholic University of Korea, Seoul, Republic of Korea
[4]Occupational and Environmental Medicine, College of Medicine, The Catholic University of Korea, Seoul, Republic of Korea

## AUTHOR ORCIDs

Raeseok Lee  http://orcid.org/0000-0002-1168-3666
Won-Bok Kim  http://orcid.org/0000-0002-9111-0365
Sung-Yeon Cho  http://orcid.org/0000-0001-5392-3405
Dong-Gun Lee  http://orcid.org/0000-0003-4655-0641

## FUNDING

| Funder | Grant(s) | Author(s) |
| --- | --- | --- |
| Ministry of Education | NRF-2022R1I1A01070887 | Raeseok Lee |

## AUTHOR CONTRIBUTIONS

Raeseok Lee, Conceptualization, Data curation, Formal analysis, Funding acquisition, Investigation, Methodology, Project administration, Writing – original draft, Writing – review and editing | Won-Bok Kim, Data curation, Formal analysis, Investigation, Methodology, Visualization, Writing – original draft, Writing – review and editing | Sung-Yeon Cho, Conceptualization, Investigation, Methodology, Project administration, Supervision, Writing – review and editing | Dukhee Nho, Data curation, Formal analysis, Methodology, Validation, Visualization, Writing – review and editing | Chulmin Park, Conceptualization, Investigation, Methodology, Project administration, Supervision, Writing – review and editing | Hye-Sun Chun, Data curation, Formal analysis, Investigation, Validation, Visualization, Writing – review and editing | Jun-Pyo Myong, Conceptualization, Investigation, Methodology, Project administration, Supervision, Writing – review and editing | Dong-Gun Lee, Conceptualization, Investigation, Methodology, Project administration, Supervision, Writing – review and editing

## DATA AVAILABILITY

All sequencing data have been included in the supplementary file. The sequence data supporting this study have been deposited in the Korean BioData Service (KBDS) with the accession number KAP241420 and can be accessed at https://kbds.re.kr/KAP241420.

## ADDITIONAL FILES

The following material is available online.

### Supplemental Material

**Figure S1 (Spectrum01902-24-s0001.tif).** Fungal colony counts and environmental conditions at each sampling site.
**Figure S2 (Spectrum01902-24-s0002.tif).** Phylogenetic tree of *Aspergillus* strains based on ITS, benA, and cyp51A gene sequences.
**Supplemental material (Spectrum01902-24-s0003.docx).** Supplemental method, tables, and figures.
**Supplemental file (Spectrum01902-24-s0004.pdf).** All results of multiple locus variable-number tandem repeat sequence types.

### Open Peer Review

**PEER REVIEW HISTORY (review-history.pdf).** An accounting of the reviewer comments and feedback.

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
