## [Reviewer comments · Microbiology Spectrum]

Microbiology Spectrum

Genetic relationships of *Aspergillus fumigatus* in hospital settings during COVID-19

Raeseok Lee, Won-Bok Kim, Sung-Yeon Cho, Dukhee Nho, Chulmin Park, Hye-Sun Chun, Jun-Pyo Myong, and Dong-Gun Lee

Corresponding Author(s): Dong-Gun Lee, The Catholic University of Korea College of Medicine

Review Timeline:

Submission Date:	July 30, 2024
Editorial Decision:	September 13, 2024
Revision Received:	October 29, 2024
Editorial Decision:	December 5, 2024
Revision Received:	January 20, 2025
Accepted:	January 31, 2025

Editor: Christina Cuomo

Reviewer(s): Disclosure of reviewer identity is with reference to reviewer comments included in decision letter(s). The following individuals involved in review of your submission have agreed to reveal their identity: Eveline Snelders (Reviewer #2)

Transaction Report:

DOI: <https://doi.org/10.1128/spectrum.01902-24>

Re: Spectrum01902-24 (Genetic relationships and transmission dynamics of *Aspergillus fumigatus* in hospital settings during COVID-19)

Dear Dr. Dong-Gun Lee:

Thank you for the privilege of reviewing your work. Below you will find my comments, instructions from the Spectrum editorial office, and the reviewer comments.

In addressing the comments from the two reviewers please note that reviewer 2 has also provided comments in an attachment. A critical point to address is that the conclusions about transmission are currently over-interpreted. VNTR data is not sufficient to make inferences about transmission; higher resolution such as from whole genome sequence would be required, or the text suggesting transmission and that the isolates are "genetically identical" (on lines 201-203 and elsewhere transmission is described) needs to be toned down. In addition, please take care to cite closely related studies in your discussion such as the Loeffert 2017 BMJ Open paper that carried out similar assessment of *Aspergillus fumigatus* in a hospital environment.

Revision Guidelines

Sincerely,
Christina Cuomo
Editor
Microbiology Spectrum

Reviewer #1 (Comments for the Author):

This manuscript entitled "Genetic relationships and transmission dynamics of *Aspergillus fumigatus* in hospital settings during COVID-19" reported the molecular epidemiology of *Aspergillus fumigatus* in the hospital in Korea during COVID-19 pandemic. COVID-19-associated pulmonary aspergillosis (CAPA) is of great concern in the world. First, the authors collected a total of 155 *A. fumigatus* strains isolated from the patients with COVID-19 and without COVID-19, and from the environments such as the hospital and outdoors. Next, they conducted the epidemiological analysis based on the molecular typing of the strains. The authors highlight the risks of CAPA with negative pressure rooms. This finding would be useful for the management of infection control in clinical settings, especially for CAPA patients. Since a secondary infection of *A. fumigatus* is life-threatening, this epidemiological study is quite important for reassessing the risk of nosocomial transmission of *A. fumigatus*. The manuscript is well written, and the data are nicely presented. I raise the minor concerns.

Minor comments

1. "*Aspergillus fumigatus*" and "*A. fumigatus*" are mixed in the text. For example, in line 32, "*Aspergillus fumigatus*" is "*A. fumigatus*" in italic.
2. "Nurse station" is "Nurse station" in Figure 2.
3. In line 178, "Fig. 3 and 4" is "Figs. 3 and 4".
4. In line 183, "Isolates exhibiting azole resistance are indicated by arrows in Figure 3." According to Supplementary File, I thought that these strains have the mutations of *cyp51A*, but are susceptible to azoles. Please correct the expression.
5. In lines 196 and 204, please correct the expression. The type of *cyp51A* of all strains in the cluster 3 is WT, but not TR mutations. TR mutations indicated TR34 and TR46.

Reviewer #2 (Comments for the Author):

This manuscript investigates the genetic relationships and transmission dynamics of *Aspergillus fumigatus* in a hospital during the COVID-19 pandemic. It compares isolates from patients with and without COVID-19 and environmental air samples, particularly in ICU patients with COVID-19. The study claims to identify potential genetic similarities based on sequence types between clinical and environmental strains, suggesting airborne contamination in negative pressure rooms. The authors state that their findings highlight the need to reassess the use of negative pressure rooms for critically ill COVID-19 patients to prevent fungal infections.

The use of 10 markers for a fungus with such high level of diversity and genome size, is not yielding reliable data, sufficient to study genome similarity or source tracing as stated by the authors. However, since all the identification and selection of isolates has already been done, adding a higher resolution method to this work, may support the proposed recommendations and conclusions.

Review attachment

This manuscript investigates the genetic relationships and transmission dynamics of *Aspergillus fumigatus* in a hospital during the COVID-19 pandemic. It compares isolates from patients with and without COVID-19 and environmental air samples, particularly in ICU patients with COVID-19. The study claims to identify potential genetic similarities based on sequence types between clinical and environmental strains, suggesting airborne contamination in negative pressure rooms. The authors state that their findings highlight the need to reassess the use of negative pressure rooms for critically ill COVID-19 patients to prevent fungal infections.

The use of 10 markers for a fungus with such high level of diversity and genome size, is not yielding reliable data, sufficient to study genome similarity or source tracing as stated by the authors. However, since all the identification and selection of isolates has already been done, adding a higher resolution method to this work, may support the proposed recommendations and conclusions.

Main points of attention

- The authors study transmission of a fungus that is highly diverse by “We applied multiple locus variable-number tandem repeat analysis to elucidate transmission pathways and relationships.” The used method will never elucidate transmission, because unless comparing clonal isolates, two unique isolates of this fungus will on average have between 5-1000 unique single nucleotide variants. Using only 10 loci (!) , will never be sufficient to determine relatedness for this fungus.
Their own results confirm this as by using only X loci, there are already 131 genotypes out of 155 isolates.
- Since their chosen MLVA method will not have sufficient power to detect relatedness in a fungus with a highly diverse genetic population, any conclusions or claims on transmission made in their manuscript cannot be supported. There can be no genetic links shown between isolates based on genotyping 10 loci only for a 30Mb genome, nor can this suggest nosocomial transmission.
- There is quite some use of unconventional phrasing of sentences, but most importantly is the lack of specificity in certain sentences. The specificity needs to improve to be able to judge if certain statements do match the available data. Additionally, sometimes data or information is not included or not referred to properly, this makes it hard to judge the accuracy of the research.
 - o Many ambiguous terms are used to describe the conclusions, such as ‘highly related’, ‘spatiotemporally relatedness’, or ‘ molecular dynamics of transmission’. Defining what this means in measurable terms seems essential to judge accuracy.
- The epidemiological analysis of the research hardly takes into account the biology of *Aspergillus fumigatus*. The fungus is ubiquitous and previous research has shown that finding genetic matches, gives limited information about the transmission or origin of an isolate. The authors should better provide this information, but also use it to make more informed conclusions about the data.

- The analysis of the VNTR is hard to follow. The clusters between Fig. 3 and Fig. 4 do not overlap, the software settings are not listed, no robustness evaluation for the MST is available, and only limited isolate numbers are shown in the phylogeny. Not only do I have doubts about the software that was used, but this also makes it impossible to judge the accuracy of certain conclusions about the putative relatedness of certain isolates.
- The authors should take into account that VNTR is only a very limited part of the genome, and gives indications of putative clonality, but this is not the same as a genotype or genetic identity.

Detailed comments

- Abstract line 32 "*Aspergillus fumigatus*" is not in italic.
- Line 39: It is hard to determine the spatiotemporality of a clinical isolate given that the moment of isolation from the patient does not have to be close to the moment in which the infection took place. How does the author take this into account when making this conclusion?
- Line 73-76: It is unclear for me what is meant by "molecular dynamics of *Aspergillus* transmission in these settings" and "acquired from the community". Community meaning outdoors, indoors?
- Line 108-110: This indicates that posaconazole was used in some of the patients, however, it does not seem to be specified in the supplementary data which of the isolates that are included in the study were isolated from these patients. I assume this data is available so why is it not included in the supplemental data?
- Line 124: Source 19 gives information about how to extract the DNA, however, there is currently no information on how the fungi were cultured. This should be included to make sure the results of this work are reproducible/comparable.
- Line 125: *cyp51A* gene was analyzed to assess its resistance profile. Yes, the TR34 and TR46 genotypes are correlated to resistance, but there are many isolates that are wild type for *cyp51A* gene and azole resistant. Apparently full MIC testing was conducted but not included here in this section. Please include at least include that MIC was done.
- Line 132, the author refers to reference 10 as multiple prior studies, but is only 1 paper. Looking up this paper, there was a referral to the method in another paper. In this paper in the discussion, it is stated: "Klassen et al have underlined the existence of possible genetic differentiation and variable recombination rates of *A. fumigatus* which could prevent correct analysis of genotyping result." This original paper of 2017 (7 years ago) seems to be rather outdated as whole genome sequencing becomes affordable and the full 30Mb genome can be assessed, rather than 10 loci. Also, please use correct original references to the method, Table S2, are these the PCR primer sequences of that reference? There is no reference mentioned, so then we would have to assumed a complete *de novo* design? Also, in the supplemental materials there is no method described for the VNTR assay.
- Line 139: PHYLOVIZ 2.0 seems to be a software specifically designed for bacterial epidemiology and surveillance. How does the software take into account both the sexual and parasexual cycles in *A. fumigatus*? For eukaryotes different analysis methods need to be used, it is a 30Mb large

genome and a species that reproduces clonal and sexually and has a very high recombination rate. Why not use splitstree?

- Line 141: Was any bootstrapping or statistics performed to assess the credibility or robustness of the MST estimations?
- Line 142: Please specify how the SDI was calculated.
- Line 149: Specify which clinical characteristics are included here, or refer to the data in the supplement where the clinical characteristics are compared.
- Line 153: Do the authors mean “colony” instead of “conidia”?
- Line 156-158: This calculation was done for the overall CFU, however, the paper is about exposure to *A. fumigatus*. Has this seasonal variation also been observed for *A. fumigatus*?
- Line 156-158: There is a variation in the CFU for different time points, however, it remains unclear how this is related to seasonality? There is only one measurement per month, there is no apparent link with the variation in humidity, and there is also a peak in May. Please elaborate on how ‘seasonal variation’ is more accurate here compared to ‘variation between measurements’.
- Line 167: Was azole resistance exactly 5% in both Covid and non-Covid patients? And where these the exact same profiles of resistance (to which azoles?), please describe in the text.
- Line 172: I highly doubt if STs are the best way to describe a *A. fumigatus* population structure. $\pm 85\%$ of the isolates are the only one in their isolate, and despite the similarities on these 10 markers, this does not mean that the isolates in one ST have the same genotype (e.g. ST43 does not even have the same MIC).
- Line 178: With the current data, it is impossible to check if these isolates are indeed closely related. There is no information about the settings of the software, the NVTR sequences, and only some isolate names are listed in the phylogeny. All I can note, is that the isolates do not even hold the same ST, whereas other isolates do belong to the same ST.
- Line 180-181: Could you define “close relationships” in a bit more biological terms? Given that there are only 10 markers used to resemble the entire genome, I think a difference of 3/10 of these markers is quite a big difference.
- Line 183-186: When examining Figure 3, three different clusters are annotated. However, there are azole-resistant isolates present in all of them? Also, there seem to be a different cluster numbers in Fig. 3 compared to Fig. 4?
- Line 186: What would be considered the main cluster?
- Line 195-197: This sentence is very unclear and I cannot understand what is meant. Also, the strains in cluster 3 do not even belong to the same ST based on 10 markers, this makes it impossible for them to be genetically identical.
- Line 201-202: There are strains that are similar, but not even to the level of ST. This means that there are potentially heaps of genetic diversity between these isolates. Additionally, how is the genetic difference between two isolates that could still make them related enough to indicate ‘intra-unit environmental transmission’ defined in this paper or broader literature?
- Line 210: ‘high genetic similarity’ in the broad sense was not found, there is ‘high genetic similarity in 10 markers’. This is a very important distinction to make, as the markers are only a proxy for the whole genome.

- Line 214-215: Nosocomial infections (patient-environment match) should only be concluded for isolates that have at least the same ST, *Cyp51a* mutations, and similar MICs. How many of these are present? Please clearly refer to this when making this statement.
- Line 231-233: The transmission of the spores was always airborne. What is new about this conclusion? Also, 'genotypes' here seems to refer to ST? A genotype is not the same as an ST.
- Line 235-236: Since the paper compared IPA with CAPA, does the *Aspergillus* environmental burden have a different effect on IPA compared to CAPA patients?
- Line 236: This conclusion focusses on CAPA patients, which is already a group with the same underlying disease. What are the individual host factors that affect acquiring hospital infections?
 - o Also, if fungal load does not seem to impact the disease development in COVID-19, why does the situation need to change?
- Line 239: A recommendation based on transmission studies with a method of poor resolution do not seem warranted at all.
- Line 248: The word 'cross-contamination' is used in an unclear manner here, what is being cross-contaminated here?
- Line 251-252: Where are these environmental or patient samples? This makes a very important distinction for these conclusions. Without specifying this, it cannot be judged if the conclusion is realistic.
- Line 260: What would be the 'multidimensional measures' that are suggested here? Same goes for 'adopting tailored strategies' in Line 276.
- Line 263: 'Culturability' is not a word. Also, *A. fumigatus* grows on a wide range of media and temperatures. Clinical isolates might be harder to culture, but here it is unspecified what isolate origin is referred to. Do they mean low culture sensitivity of clinical isolates instead?
- Line 270: It is very unclear what is meant by 'stringent environmental controls may have reduced the culturability of the *Aspergillus*'.

Detailed comments 'Supplementary methods'

To make the reviewing easier, line numbers were added 'continuous' from the methods section onwards.

- Line 16: The colonies of the plates, do these originate from the subculture or the original culture? Also, please list the culturing conditions.
- Line 20: The PCR was performed under 'specific conditions'. Please define the conditions.
- Line 28: Given that the MIC is defined here as a reduction of 50% fungal growth. Do you mean a MIC₅₀? Also, at what azole concentration did you consider an isolate to be resistant? Which breakingpoints are used?
- Line 30: Please define 'abnormal hyphal growth'. If the format allows, this could also be done by pictures.
- Line 37: Please list the company that performed the sequencing, type of sequencing etc.

Response to Editor's and Reviewers Comments : Manuscript ID [Spectrum01902-24]

We would like to thank the editor and reviewers for the invaluable comments that helped a great deal in improving the overall quality of our original manuscript. Please find attached our revised paper and below our point-by-point responses to your insightful suggestions and valuable comments.

Editor

A critical point to address is that the conclusions about transmission are currently over-interpreted. VNTR data is not sufficient to make inferences about transmission; higher resolution such as from whole genome sequence would be required, or the text suggesting transmission and that the isolates are "genetically identical" (on lines 201-203 and elsewhere transmission is described) needs to be toned down. In addition, please take care to cite closely related studies in your discussion such as the Loeffert 2017 BMJ Open paper that carried out similar assessment of *Aspergillus fumigatus* in a hospital environment.

RESPONSE (RS): We appreciate your valuable insights. Although the STR method in fungal molecular epidemiology has certain limitations, as highlighted by the reviewers, it offers considerable advantages in terms of time efficiency, cost-effectiveness, and accessibility. Prior studies have shown that this method provides high resolution, establishing its importance as a useful tool in fungal epidemiology research. However, since this method predominantly identifies associations rather than providing conclusive evidence of transmission, we have adjusted the language throughout the manuscript to reflect this distinction. Furthermore, following suggestions from the previous office review, we have removed the term "transmission" from the title to ensure clarity and prevent any potential misinterpretation of the study's scope and significance. Please refer to Title and Abstract lines 27-29, 41-44, Introduction lines 79-83, Results lines 180, 189-191, 197, 204-207. Discussion lines 219, 221-222, 224-228, 257-262.

Reviewer 1

This manuscript entitled "Genetic relationships and transmission dynamics of *Aspergillus fumigatus* in hospital settings during COVID-19" reported the molecular epidemiology of *Aspergillus fumigatus* in the hospital in Korea during COVID-19 pandemic. COVID-19-associated pulmonary aspergillosis (CAPA) is of great concern in the world. First, the authors collected a total of 155 *A. fumigatus* strains isolated from the patients with COVID-19 and without COVID-19, and from the environments such as the hospital and outdoors. Next, they conducted the epidemiological analysis based on the molecular typing of the strains. The authors highlight the risks of CAPA with negative pressure rooms. This finding would be useful for the management of infection control in clinical settings, especially for CAPA patients. Since a secondary infection of *A. fumigatus* is life-threatening, this epidemiological study is quite important for reassessing the risk of nosocomial transmission of *A. fumigatus*. The manuscript is well written, and the data are nicely presented. I raise the minor concerns.

Minor comments

1. "*Aspergillus fumigatus*" and "*A. fumigatus*" are mixed in the text. For example, in line 32, "*Aspergillus fumigatus*" is "*A. fumigatus*" in italic.

RESPONSE (RS): Thank you for your valuable feedback. In the first mention, we have used the full name, and for all subsequent mentions, we have consistently used the abbreviation *A. fumigatus*. Please refer to lines 31 and 78 for these changes.

2. "Nurse statation" is "Nurse station" in Figure 2.

RESPONSE (RS): Thank you for your detailed review. we have corrected the typo in the corresponding Figure. Please refer to Figure 1 as formally figure 2.

3. In line 178, "Fig. 3 and 4" is "Figs. 3 and 4".

RESPONSE (RS): Thank you for your meticulous review. We have revised the content accordingly. Please refer to line 187.

4. In line 183, "Isolates exhibiting azole resistance are indicated by arrows in Figure 3." According to Supplementary File, I thought that these strains have the mutations of *cyp51A*, but are susceptible to azoles. Please correct the expression.

RESPONSE (WB): Thank you for your detailed feedback. As you pointed out, we recognized that Figure 3 lacked sufficient explanation, potentially leading to confusion between azole-resistant and susceptible isolates. Therefore, we have revised both the main text and the figure legend. Please refer to line 192-193 and line 486-492.

5. In lines 196 and 204, please correct the expression. The type of *cyp51A* of all strains in

the cluster 3 is WT, but not TR mutations. TR mutations indicated TR34 and TR46.

RESPONSE (WB): Thank you for your detailed feedback. As per your suggestions, we have revised our previous incorrect expression to the following statement. Please refer to line 204-207 line 211-214.

Reviewer 2

This manuscript investigates the genetic relationships and transmission dynamics of *Aspergillus fumigatus* in a hospital during the COVID-19 pandemic. It compares isolates from patients with and without COVID-19 and environmental air samples, particularly in ICU patients with COVID-19. The study claims to identify potential genetic similarities based on sequence types between clinical and environmental strains, suggesting airborne contamination in negative pressure rooms. The authors state that their findings highlight the need to reassess the use of negative pressure rooms for critically ill COVID-19 patients to prevent fungal infections. The use of 10 markers for a fungus with such high level of diversity and genome size, is not yielding reliable data, sufficient to study genome similarity or source tracing as stated by the authors. However, since all the identification and selection of isolates has already been done, adding a higher resolution method to this work, may support the proposed recommendations and conclusions.

RESPONSE (RS): Thank you for your detailed insights. After thorough discussion among all authors, we agreed with many of your points. In particular, given that STR analysis (VNTR analysis) is a core methodology of this study, we have clearly outlined its advantages and limitations, adjusting the tone of the expressions and revising the wording for greater clarity. However, we kindly ask for your understanding regarding the lack of further molecular analyses due to time constraints, limited samples, and resource availability.

Main points of attention

- The authors study transmission of a fungus that is highly diverse by “We applied multiple locus variable-number tandem repeat analysis to elucidate transmission pathways and relationships.” The used method will never elucidate transmission, because unless comparing clonal isolates, two unique isolates of this fungus will on average have between 5-1000 unique single nucleotide variants. Using only 10 loci (!) , will never be sufficient to determine relatedness for this fungus. Their own results confirm this as by using only X loci, there are already 131 genotypes out of 155 isolates.

RESPONSE (RS): Thank you for your insightful comments. We fully agree that proving all transmission events using only 10 loci is challenging. However, the use of STR assays in molecular epidemiology has been well established in bacteria for a long time and, more recently, has also been validated for fungi, where it is known for its high discriminatory power. That said, as you pointed out, this method provides indirect evidence rather than definitive proof. We have revised the title and text to tone down the language accordingly, and further included additional remarks in the limitations section to aid readers in their interpretation and understanding. Please refer to Title and Abstract lines 27-29, 41-44, Introduction lines 79-83, Results lines 180, 189-191, 197, 204-207. Discussion lines 219. 221-222, 224-228, 257-262.

- Since their chosen MLVA method will not have sufficient power to detect relatedness in a fungus with a highly diverse genetic population, any conclusions or claims on transmission made in their manuscript cannot be supported. There can be no genetic links shown between isolates based on genotyping 10 loci only for a 30Mb genome, nor

can this suggest nosocomial transmission.

RESPONSE (WB): According to previous studies, the STR method (VNTR assay) is known for its strong ability to detect genetic diversity and relatedness. Analyzing all genes through NGS or whole genome sequencing (WGS) faces limitations in terms of time, sample availability, and resources. To overcome these limitations, many studies have employed the STR method to investigate molecular diversity. In this context, the MLVA method was developed, offering a more precise diversity analysis by targeting longer repeat sequences rather than short ones. This method has been recognized as useful for assessing hospital environments, as acknowledged by the *Clinical Infectious Diseases* (CID) journal.

In our study, we referenced this analytical method to assess the molecular epidemiological relatedness between hospital environmental samples and patient-derived isolates. While it is logically undeniable that WGS offers a more accurate comparison of genetic similarity, numerous studies and papers have demonstrated that the STR method is validated and reproducible. In resource-limited situations, we believe it is also undeniable that the STR method provides a faster and more efficient means of genetic comparison than WGS.

- There is quite some use of unconventional phrasing of sentences, but most importantly is the lack of specificity in certain sentences. The specificity needs to improve to be able to judge if certain statements do match the available data. Additionally, sometimes data or information is not included or not referred to properly, this makes it hard to judge the accuracy of the research. Many ambiguous terms are used to describe the conclusions, such as 'highly related', 'spatiotemporally relatedness', or 'molecular dynamics of transmission'. Defining what this means in measurable terms seems essential to judge accuracy.

RESPONSE (RS): We thoroughly reviewed the manuscript and revised the wording to ensure it conveys a clearer understanding to readers. Please refer to Title, Abstract lines 27-29, 37-39 Introduction lines 74-75, Results lines 197, 204-207, 210-211, and Figure legend line 503 for the changes.

- The epidemiological analysis of the research hardly takes into account the biology of *Aspergillus fumigatus*. The fungus is ubiquitous and previous research has shown that finding genetic matches, gives limited information about the transmission or origin of an isolate. The authors should better provide this information, but also use it to make more informed conclusions about the data.

RESPONSE (RS): We sincerely appreciate your detailed and constructive comments. As you mentioned, although this study aimed to overcome the limitations of previous studies, we recognize, as previously stated, that the inherent limitations of MLVA prevent us from drawing fully robust conclusions. Following the previous responses, we have revised the manuscript by toning down the language and adding more details in the limitations section to enhance the readers' understanding. Please refer to Title and Abstract lines 27-29, 41-44, Introduction lines 79-83, Results lines 180, 189-191, 197, 204-207. Discussion lines 219, 221-222, 224-228, 257-262.

- The analysis of the VNTR is hard to follow. The clusters between Fig. 3 and Fig. 4 do not overlap, the software settings are not listed, no robustness evaluation for the MST is available, and only limited isolate numbers are shown in the phylogeny. Not only do I have doubts about the software that was used, but this also makes it impossible to judge the accuracy of certain conclusions about the putative relatedness of certain isolates.

RESPONSE (WB): Thank you for your valuable feedback. We overlooked the ambiguity in the cluster descriptions between Figs. 3 and 4. Consequently, we have labeled Fig. 3 as the cluster, while Fig. 4 has been revised to “the clade”. This adjustment has been reflected in the figure legends and the main text. Additionally, the settings used in the software are described in the Materials & Methods section, Lines 139 to 143. This free tool is easily accessible and is comparable to paid programs like Bionumerics. Moreover, detailed usage instructions for this tool are available on its website.

The VNTR method was performed in triplicate, and the same STR results were confirmed in all experiments. Please refer to lines 137-143.

The number of isolates shown in Fig. 4 was derived from the total phylogenetic tree generated using the program, focusing on the group that includes isolates from COVID-19 patients, as mentioned in the text (Lines 198-200), with a 1.5 cut-off for revising the phylogenetic tree.

The topic and goal of this paper were to organize the phylogenetic relationships of specific isolates of *A. fumigatus* causing CAPA using the STR (VNTR) method. Based on this, we aimed to focus on whether there is an environmental association with the origin of the isolates causing CAPA.

- The authors should take into account that VNTR is only a very limited part of the genome, and gives indications of putative clonality, but this is not the same as a genotype or genetic identity.

RESPONSE (RS): Thank you again for your meticulous comments. We fully acknowledge your points regarding the limitations of VNTR. As mentioned earlier, these limitations have been discussed in the relevant section, and the overall tone of the manuscript has been adjusted to assist readers in interpreting and evaluating the data.

Detailed comments

- **Abstract line 32 “*Aspergillus fumigatus*” is not in italic.**

RESPONSE (RS): Edited to be in italics. **Please refer Abstract line 31.**

- **Line 39: It is hard to determine the spatiotemporality of a clinical isolate given that the moment of isolation from the patient does not have to be close to the moment in which the infection took place. How does the author take this into account when making this conclusion?**

RESPONSE (RS): Thank you for your detailed comments. The term "spatiotemporal" was used to describe the genetic linkage observed over a certain period in the same COVID-19 ICU unit. However, We agree that this term might be somewhat ambiguous and subject to subjective interpretation. We have revised the wording to convey the meaning more clearly. Please refer to Abstract, lines 37-39.

- **Line 73-76: It is unclear for me what is meant by “molecular dynamics of *Aspergillus* transmission in these settings” and “acquired from the community”. Community meaning outdoors, indoors?**

RESPONSE (RS): Thank you for your careful feedback. We have revised the ambiguous terminology related to "molecular dynamics." The terms "community-acquired" vs. "hospital-acquired" infections are widely used in accordance with CDC guidelines and numerous prior studies. For hospital-acquired infections (HAI), we used the standard definition of infections that typically occur after hospitalization and manifest 48 h after admission, while cases with positive cultures upon admission were classified as community-acquired. Please refer lines 74-75.

- **Line 108-110: This indicates that posaconazole was used in some of the patients, however, it does not seem to be specified in the supplementary data which of the isolates that are included in the study were isolated from these patients. I assume this data is available so why is it not included in the supplemental data?**

RESPONSE (RS): Thank you for your comments. There were no patients receiving Posaconazole prophylaxis in the COVID-19 cohort. However, we have added details on patients who received Posaconazole prophylaxis in the non-CAPA group to the supplementary data, so readers and future researchers can easily access this information. Please refer the supplementary file.

- **Line 124: Source 19 gives information about how to extract the DNA, however, there is currently no information on how the fungi were cultured. This should be included to make sure the results of this work are reproducible/comparable.**

RESPONSE (WB): The strains obtained through air sampling were cultured on Sabouraud dextrose agar (SDA) at 35°C for 4–5 days, as described in the Supplementary Methods. In contrast, the clinical strains received underwent a separate culture process, after which only

the conidia of the strains suspected to be *A. fumigatus* were transferred; therefore, it is challenging to confirm the specific culture conditions for those strains. The transferred conidia were cultured in the same manner as the air samples. This information has been updated, and further details can be found in the supplementary file.

- Line 125: cyp51A gene was analyzed to assess its resistance profile. Yes, the TR34 and TR46 genotypes are correlated to resistance, but there are many isolates that are wild type for cyp51A gene and azole resistant. Apparently full MIC testing was conducted but not included here in this section. Please include at least include that MIC was done.

RESPONSE (WB): Thank you for your valuable comments. The MIC method is described in the Supplementary Methods. Additionally, a sentence related to the MIC analysis has been added to the main text, and the detailed methods have been included in the supplementary file as before. Please refer to lines 126-127.

- Line 132, the author refers to reference 10 as multiple prior studies, but is only 1 paper. Looking up this paper, there was a referral to the method in another paper. In this paper in the discussion, it is stated: “Klassen et al have underlined the existence of possible genetic differentiation and variable recombination rates of *A. fumigatus* which could prevent correct analysis of genotyping result.” This original paper of 2017 (7 years ago) seems to be rather outdated as whole genome sequencing becomes affordable and the full 30Mb genome can be assessed, rather than 10 loci. Also, please use correct original references to the method, Table S2, are these the PCR primer sequences of that reference? There is no reference mentioned, so then we would have to assumed a complete de novo design? Also, in the supplemental materials there is no method described for the VNTR assay.

RESPONSE (WB):

Thank you for your valuable comments. In addition to one representative paper, we have included an additional reference from the original BMJ paper, adding the following references in Line 134: [20] doi: 10.1136/bmjopen-2017-018109, [21] 10.1016/j.mycmed.2018.05.007, [22] 10.1186/1471-2180-10-315, and [23] 10.3382/ps.2013-03541.

Recent papers have shown a trend of requiring not only MIC analysis results for *Aspergillus* strains isolated from clinical and environmental samples but also STR analysis data and sequencing DNA information. The potential for genetic differentiation and variable recombination rates mentioned in the 2017 paper differs by species and can be influenced by subculturing or environmental factors. One specific study reported an increase of one TR in one of the nine loci after 50 to 100 rounds of subculturing using a different target STR method (reference : 10.3389/fcimb.2019.00082).

While the cost of WGS (whole-genome sequencing) is decreasing, it remains a method with limited accessibility in several countries, including developing nations. Furthermore, WGS incurs a minimum cost of approximately \$1,500 per sample, and analyzing dozens to hundreds of samples simultaneously requires high-performance servers and software. We believe that accurate genomic analysis and species epidemiological analysis can only be achieved by comparing all samples simultaneously.

The PCR primer sequences used in Table S2 were designed in the 2017 paper. As this study was an extension of research published by the same authors in 2019, we referenced only the 2019 paper; however, we have also included the 2017 paper.

- Line 139: PHYLOVIZ 2.0 seems to be a software specifically designed for bacterial epidemiology and surveillance. How does the software take into account both the sexual and parasexual cycles in *A. fumigatus*? For eukaryotes different analysis methods need to be used, it is a 30Mb large genome and a species that reproduces clonal and sexually and has a very high recombination rate. Why not use splitree?

RESPONSE (WB): Thank you for the valuable information. We confirmed that the Splitree program, developed in 2006, is specialized in phylogenetic analysis. The reason we used the PHYLOVIZ 2.0 program is that it has been employed in many STR analysis papers, including those we referenced. In contrast, it seems that Splitree is not widely recognized, as we did not find any usage cases in the fungal papers we referred to regarding STR analysis and MST diagrams. However, upon a general review, we learned that Splitree is a very useful program.

In our study, we did not analyze the entire 30 Mb of genes; instead, we focused on the *benA* gene for species classification and performed analyses on some VNTR loci. Therefore, we did not consider sexual and asexual reproduction. Clinically, *A. fumigatus* is known to reproduce asexually (reference: 10.3390/jof7080599), and we emphasized the possibility of identifying whether the isolates found in clinical or environmental settings were indeed *A. fumigatus* and their potential environmental origins. To confirm whether sexual or asexual reproduction occurs, the presence of MAT genes and RNA expression must be analyzed; however, we deemed such methods unsuitable for clinical application.

It is true that *A. fumigatus* has a rapid rate of genetic recombination, and genetic exchanges, fusions, and losses can occur with every cell division across all populations. In most cases, these changes are repaired through gene repair mechanisms, but a few may become modified, particularly in intron regions. The fingerprinting method through STR analysis shows diversity in STs, whereas genetic analysis via WGS is likely to reveal individual STs for each isolate. Therefore, if genetic diversity is confirmed to be similar through STR analysis across some loci, and if the rapid genetic recombination results in numerous mutations, this similar genetic diversity may serve as significant evidence that the isolates originated from the environment and increased the likelihood of opportunistic infections in patients.

- Line 141: Was any bootstrapping or statistics performed to assess the credibility or robustness of the MST estimations?

RESPONSE (WB): Thank you for your valuable question. In our study, we did not perform bootstrapping to evaluate the reliability and robustness of the Minimum Spanning Tree (MST) estimation. While MST analysis is useful for understanding the distribution and structure of data, we primarily utilized it as a visual representation of the existing STR analysis results to align with the goals of this research. However, in future studies, we plan to consider such statistical methods to enhance the reliability of MST estimations. We agree that it is important to introduce additional analytical methodologies to assess the reliability of MST.

- Line 142: Please specify how the SDI was calculated.

RESPONSE (WB): Thank you for your detailed comments. The Simpson diversity index (SDI) is calculated to evaluate the polymorphic ability of microsatellite markers to distinguish between isolated strains. It is calculated as follows. Most STR analysis tools automatically compute the SDI value.

ST	Absolute Frequency	Relative Frequency
----	--------------------	--------------------

$$D = 1 - \frac{1}{N(N-1)} \sum_{j=1}^s x_j(x_j - 1)$$

N is the number of isolates, and x_j is the proportion of isolates belonging to the j-th group of TR repetitions.

- Line 149: Specify which clinical characteristics are included here, or refer to the data in the supplement where the clinical characteristics are compared.

RESPONSE (RS): Thank you for your careful feedback. The detailed characteristics are thoroughly described in Table 1 and also referenced in the main text. To avoid redundancy, only the key characteristics are explained in the text. Please refer to lines 154-157 and Table 1.

- Line 153: Do the authors mean “colony” instead of “conidia”?

RESPONSE (WB): Thank you for your insightful observation. We have decided that using the term 'isolates' instead of 'conidia' is more appropriate. Each isolate is regarded as a single conidium. We will revise the manuscript to standardize the use of '**isolates**' and eliminate any confusion regarding the terms 'colony' and 'conidia'. Please refer to line 159.

- Line 156-158: This calculation was done for the overall CFU, however, the paper is about exposure to *A. fumigatus*. Has this seasonal variation also been observed for *A. fumigatus*?

RESPONSE (WB): Thank you for your detailed feedback. As mentioned in a previous question, a single conidia of *A. fumigatus* forms one colony. Therefore, a single conidia is equivalent to one colony. In Figure S1, you can observe the seasonal variations and current status of *A. fumigatus*. *A. fumigatus* appeared in all seasons without significant differences (spring: 36 isolates, summer: 42 isolates, fall: 48 isolates, winter: 29 isolates). We will add a discussion on seasonal variability to the main text. Please refer to lines 161-165

- Line 156-158: There is a variation in the CFU for different time points, however, it remains unclear how this is related to seasonality? There is only one measurement per month, there is no apparent link with the variation in humidity, and there is also a peak in May. Please elaborate on how ‘seasonal variation’ is more accurate here compared to ‘variation between measurements’.

RESPONSE (RS): Thank you for your thoughtful comments. The correlation between humidity and fungal burden is well-known, and thus, a year-long observation is typically recommended when studying fungal burden, rather than focusing on a single time point. While high CFU counts were observed outdoors during May, the overall trend of humidity and outdoor CFU changes followed a similar pattern (Fig S1). In contrast, in hospital rooms equipped with HEPA filters, the differences were minimized. However, since this observation has not been statistically validated and outliers are present, I have removed the term "significant" and revised the text to help readers interpret the findings, as suggested. Please refer lines 161-166.

- Line 167: Was azole resistance exactly 5% in both Covid and non-Covid patients? And where these the exact same profiles of resistance (to which azoles?), please describe in the text.

RESPONSE (RS): The phrase "approximately 5%" was used in the sentence, and more detailed information is provided in Table 2. To minimize redundancy, the general text avoids repeating the same details. In the COVID-19 group, the rate was 5.3%, while in the non-COVID-19 group, it was 5.7%, with corresponding MICs for each azole listed in Table 2. Please refer Table 2.

- Line 172: I highly doubt if STs are the best way to describe a *A. fumigatus* population structure. ±85% of the isolates are the only one in their isolate, and despite the similarities on these 10 markers, this does not mean that the isolates in one ST have the same genotype (e.g. ST43 does not even have the same MIC).

RESPONSE (WB): Thank you for your valuable comments. We are aware that there is a method for describing population structure by individually sequencing specific target genes (such as ITS, *benA*, *Calmodulin*, etc., around 5–10 genes), concatenating them, and comparing genetic diversity to construct a phylogenetic tree. However, in our study, we were able to establish about 85% of individual Sequence Types (STs) through Short Tandem Repeat (STR) analysis, which is a relatively straightforward and rapid method.

Whole Genome Sequencing (WGS) is more expensive and time-consuming, and We believe that using this method for gene comparison might actually lower similarity. For instance, genes acquired by *A. fumigatus* after opportunistic infections in patients may be difficult to distinguish using WGS. The purpose of this paper is to emphasize the importance of environmental management for infection prevention due to the genetic similarity of *A. fumigatus* genotypes found in clinical and environmental settings.

ST43 consists of a single isolate, while the similar ST93 consists of three isolates. The Minimum Inhibitory Concentration (MIC) values may vary by 1–2 fold depending on the experimenter's proficiency or minor differences. However, a difference between 0.25 and 0.06 does not significantly affect azole susceptibility and can be considered practically the same.

- Line 178: With the current data, it is impossible to check if these isolates are indeed closely related. There is no information about the settings of the software, the NVTR sequences, and only some isolate names are listed in the phylogeny. All I can note, is that the isolates do not even hold the same ST, whereas other isolates to belong to the same ST.

RESPONSE (WB): Thank you for your detailed feedback. The content has already been addressed in previous responses. Regarding the software settings, we used the default values set in PHYLOVIZ 2.0, and as mentioned in the text, the Minimum Spanning Tree (MST) analysis was conducted using the Euclidean and goeBURST distance methods. No additional settings were applied.

Sequence information for the VNTRs is included in the reference literature, and the variability is expressed based on the degree of repetition of the same gene sequence according to changes in the tandem repeat (TR) values, so we did not include additional details. Furthermore, the phylogenetic tree was created to examine the similarity among CAPA isolates, so the CAPA isolates were separately extracted and illustrated in the schematic."

- Line 180-181: Could you define "close relationships" in a bit more biological terms? Given that there are only 10 markers used to resemble the entire genome, I think a difference of 3/10 of these markers is quite a big difference.

RESPONSE (RS): Thank you for your insightful feedback. As previously mentioned, while the MLVA method has different limitations compared to NGS, it remains a highly accessible, cost-effective, and highly discriminatory molecular fingerprinting tool. In VNTR analysis, a difference of 1-2 markers typically indicates high relatedness, and some studies suggest that even up to 3 markers can still indicate genetic linkage [J Fungi (Basel). 2023 Feb 24;9(3):298.]. However, as you pointed out, the use of the term "close" might lead to misinterpretation, so we have revised the wording accordingly. Please refer lines 189-191.

- Line 183-186: When examining Figure 3, three different clusters are annotated. However, there are azole-resistant isolates present in all of them? Also, there seem to be a different cluster numbers in Fig. 3 compared to Fig. 4?

RESPONSE (WB): Thank you for your detailed feedback. The arrows and labeling indicated in Figure 3 correspond to the CAPA isolates, and the presence or absence of TR mutations shown behind indicates the azole-resistant isolates. We have reflected this in the updated figure legend. Below is the revised Figure 3 legend:

Figure 3. The tree features 131 genotypes (circles) derived from 155 isolates. Circles are color-coded to indicate sources: red for isolates from patients with COVID-19, light green for isolates from patients without COVID-19, and blue for environmental samples. Circle sizes indicate the number of isolates sharing the same genotype. Each line represents the distance between genotypes expressed differently depending on the number of differences. Bold solid line - 1 marker, solid line - 2 markers, dotted line - 3 markers. TR, azole-resistant with tandem repeat. Arrow + red circle + red label: CAPA with azole susceptible; Arrow + red circle + red label + TR: CAPA with azole-resistant; Arrow + non-red circle + black label +

TR: non-CAPA with azole-resistant

- Line 186: What would be considered the main cluster?

RESPONSE (RS): The term "cluster" referred to the resistant isolates discussed in the previous sentence. However, recognizing that this expression could be unclear to readers, we have revised the cluster to clade for clarity. Please refer lines 198-215.

- Line 195-197: This sentence is very unclear and I cannot understand what is meant. Also, the strains in cluster 3 do not even belong to the same ST based on 10 markers, this makes it impossible for them to be genetically identical.

RESPONSE (WB): Thank you for your detailed feedback. As we mentioned earlier, a difference of 1-2 markers in VNTR analysis is typically interpreted as indicating genetic relatedness. The expression that they must be genetically identical does not align with genetic relatedness.

Furthermore, in the case of bacteria that reproduce solely through asexual methods such as binary fission or budding, STR analysis tends to show about 90% of distinct sequence types (STs). In bacteria, the number of markers is significantly higher than in fungi, and even if there are many differing markers, they are still considered similar within a certain cut-off range.

- Line 201-202: There are strains that are similar, but not even to the level of ST. This means that there are potentially heaps of genetic diversity between these isolates. Additionally, how is the genetic difference between two isolates that could still make them related enough to indicate 'intra-unit environmental transmission' defined in this paper or broader literature?

RESPONSE (RS): Thank you for your detailed comments. As noted earlier, in VNTR analysis, a difference of 1-2 markers is generally interpreted as indicating genetic relatedness. In this study, clades 12-13 in Figure 4 show differences of no more than 2 markers, suggesting high relatedness rather than identity, and were interpreted in the context of spatial and temporal linkage. However, acknowledging that this is based on possible analysis outcomes, we have toned down the language to avoid any misunderstanding. Please refer lines 210-211.

- Line 210: 'high genetic similarity' in the broad sense was not found, there is 'high genetic similarity in 10 markers'. This is a very important distinction to make, as the markers are only a proxy for the whole genome.

RESPONSE (RS): As previously stated, although we did not perform whole-genome sequencing, the VNTR method, based on substantial scientific evidence, has shown that relatedness can be confirmed with differences of 1–2, and up to 3, markers. We have added additional details on the strengths and limitations of the VNTR method and toned down the relevant sentences as suggested. Please refer line 219.

- Line 214-215: Nosocomial infections (patient-environment match) should only be concluded for isolates that have at least the same ST, Cyp51a mutations, and similar MICs. How many of these are present? Please clearly refer to this when making this statement.

RESPONSE (WB): Thank you for your detailed feedback. As mentioned earlier, genetic recombination in fungi occurs rapidly. Fungi that invade the human body as opportunistic infections may undergo modifications in certain genes during the process of adaptation or evasion. Additionally, during the proliferation process, there may be 1–2 additional TR repeats, similar to what was mentioned in the references, and this is also considered when judging genetic relatedness.

External genetic changes can affect phenotypes; however, an increase in TR counts in the intron regions does not significantly impact the phenotype or expression of fungi. We determined genetic relatedness based solely on ST, and the results of *cyp51A* or MIC analyses were added for reference in the species analysis. Therefore, since the results of *cyp51A* gene analysis or MIC analysis are not included in the STR analysis, we believe that the existing clustering patterns will remain unchanged.

- Line 231-233: The transmission of the spores was always airborne. What is new about this conclusion? Also, ‘genotypes’ here seems to refer to ST? A genotype is not the same as an ST.

RESPONSE (RS): Thank you for your detailed feedback. As you pointed out, the airborne transmission of *Aspergillus* spores is well-established. However, as mentioned throughout the introduction and discussion, there has been debate over whether CAPA infections are due to community-acquired *Aspergillus* (where the fungus was already colonizing the patient) or nosocomial transmission influenced by negative pressure rooms. This distinction is critical not only for the treatment of COVID-19 but also for future pandemics involving respiratory viruses, as it will influence decisions about patient care environments. By confirming the possibility of nosocomial transmission in this study, we have provided additional evidence to address this unmet need and contribute to patient safety through more informed decisions about treatment environments. We revised the sentences to clarify the study's significance for readers. Please refer to lines 240-241.

- Line 235-236: Since the paper compared IPA with CAPA, does the *Aspergillus* environmental burden have a different effect on IPA compared to CAPA patients?

RESPONSE (RS): Thank you for your important comment. To date, we have not found studies that distinguish between the effects of environmental *Aspergillus* burden on non-CAPA versus CAPA patients. While some studies have noted an increase in invasive pulmonary aspergillosis (IPA) associated with higher environmental *Aspergillus* burdens, particularly during construction, no research has specifically examined this effect in CAPA patients.

- Line 236: This conclusion focusses on CAPA patients, which is already a group with the same underlying disease. What are the individual host factors that affect acquiring hospital infections? Also, if fungal load does not seem to impact the disease development in COVID-19, why does the situation need to change?

RESPONSE (RS): Thank you for your insightful feedback. As mentioned in the discussion, the severity of the patient's condition (in our study, critically ill patients on mechanical ventilation due to high oxygen demands) is a significant factor in CAPA development. This is consistent with previous epidemiological studies, which explain that mucosal damage facilitates *Aspergillus* colonization and invasion. Therefore, as we noted in the text, even without a direct effect from environmental burden, there is clear evidence that nosocomial airborne transmission can occur. For high-risk patients, it may be necessary to consider treatment in a neutral environment, as opposed to automatically defaulting to negative pressure rooms, as is currently recommended. This is the argument we propose in our discussion. Please refer to lines 257-259.

- Line 239: A recommendation based on transmission studies with a method of poor resolution do not seem warranted at all.

RESPONSE (RS): Thank you for your comments. We fully agree that the VNTR method has its limitations. However, as noted earlier, VNTR-based molecular fingerprinting is widely recognized in fungal research, and despite its limitations compared to whole-genome sequencing, it remains a valuable scientific tool due to its accessibility, cost-efficiency, and high discriminatory power. After careful consideration, we concluded that it is important to present the possibility of nosocomial transmission based on spatial and temporal relatedness, as this remains a critical issue for patient care in the next pandemic. We appreciate your understanding of this approach.

- Line 248: The word 'cross-contamination' is used in an unclear manner here, what is being cross-contaminated here?

RESPONSE (RS): Cross-contamination is generally defined as "the physical movement or transfer of harmful pathogens from one person, object, or place to another," and we used it in the same context in this study. Genetically similar strains with identical resistance patterns appeared at different times in a room with six air exchanges per hour, suggesting environmental contamination rather than airborne transmission. We toned down the language to avoid overinterpretation and help readers understand the findings more clearly. Please refer to lines 259-262.

- Line 251-252: Where this environmental or patient samples? This makes a very important distinction for this conclusions. Without specifying this, it cannot be judged if the conclusion is realistic.

RESPONSE (RS): The samples collected from patients and the environment are detailed in Figure 4. Except for one eye discharge sample, all patient samples were respiratory.

Environmental samples were air-captured from the COVID-19 treatment rooms, hallways, and outdoor hospital areas, as described in the methods section. However, we were unable to conduct sampling of surfaces within the hospital rooms, making it difficult to directly prove transmission via environmental contamination. The wording has been revised and toned down accordingly to reflect this limitation. Please refer to lines 257-259.

- Line 260: What would be the ‘multidimensional measures’ that are suggested here? Same goes for ‘adpoting tailored strategies’ in Line 276.

RESPONSE (RS): During the COVID-19 pandemic, infection control efforts have often focused solely on negative pressure ventilation systems. However, it is crucial to go beyond this approach and develop infection control guidelines that also consider the ventilation system, environmental cleaning, and the underlying health conditions of the patients. These factors are essential to minimizing the occurrence of complications such as CAPA in critically ill patients. Based on your feedback, we have revised the sentences to clarify this point. Please refer to lines 270-271.

- Line 263: ‘Culturability’ is not a word. Also, *A. fumigatus* grows on a wide range of media and temperatures. Clinical isolates might be harder to culture, but here it is unspecific what isolate origin is referred to. Do they mean low culture sensitivity of clinical isolates instead?

RESPONSE (RS): Thank you for your detailed feedback and comments. As you mentioned, the sentence referred to the low sensitivity of *Aspergillus* cultures from clinical samples. To make this clearer, we have revised the sentence using the phrasing you suggested. Please refer lines 270-271 and 274-275.

- Line 270: It is very unclear what is meant by ‘stringent environmental controls may have reduced the culturability of the *Aspergillus*’.

RESPONSE (RS): This study emphasizes that the low positivity rate of *Aspergillus* cultures from air samples is due to the strict implementation of a rapid ventilation system and enhanced environmental cleaning in the specialized COVID-19 ICU units. As noted in your comments, the original expression was somewhat ambiguous. Therefore, we have revised the sentence to be more specific and clear. Please refer to lines 281-283.

Detailed comments ‘Supplementary methods’

To make the reviewing easier, line numbers were added ‘continious’ from the methods section onwards.

- Line 16: The colonies of the plates, do these originate from the subculture or the original culture? Also, please list the culturing conditions.

RESPONSE (WB): Thank you for your detailed feedback and comments. The term 'colony'

mentioned in this sentence originated from the original culture. The original culture medium was derived from air samples collected from the environment, in which many species of fungi were mixed and cultured. Therefore, to conduct subculturing at the species level, we diluted the samples to the single colony level before performing the subculture.

This process clarifies that subculturing was conducted from the original culture medium to the single colony level. Additionally, as previously explained, the culturing conditions were carried out on SDA medium at 35 degrees for 4-5 days, so we did not add this separately.

- Line 20: The PCR was performed under ‘specific conditions’. Please define the conditions.

RESPONSE (WB): Thank you for your detailed feedback and comments. The PCR conditions were carried out according to the references used in the main text of the paper, and related details have been added to Table S1.

- Line 28: Given that the MIC is defined here as a reduction of 50% fungal growth. Do you mean a MIC50? Also, at what azole concentration did you consider an isolate to be resistant? Which breaking points are used?

RESPONSE (WB): The criteria and calculation method for a 50% reduction in MICs are included in the relevant references. Different resistance criteria are applied for each azole and all antifungals. For example, for posaconazole, resistance is determined when the growth inhibition concentration is 1 or higher. The growth inhibition concentration is defined based on the degree of growth inhibition observed under a microscope at various concentrations.

- Line 30: Please define ‘abnormal hyphal growth’. If the format allows, this could also be done by pictures.

RESPONSE (WB): Abnormal hyphal growth in the MEC refers to the hyphae growing in a fragmented and clumped manner, which indicates suppressed growth. The echinocandin-class drugs used in the MEC analysis target fungal cell walls by inhibiting beta-D-glucan synthesis, which is why such abnormal growth forms are observed.

Reference : 10.1093/jac/dkaa102

- Line 37: Please list the company that performed the sequencing, type of sequencing etc.

RESPONSE (WB): Sequencing was performed by the same company that conducted the STR analysis, ensuring consistency in methodology across our genomic assessments. Please refer to line 37.

Re: Spectrum01902-24R1 (Genetic relationships of *Aspergillus fumigatus* in hospital settings during COVID-19)

Dear Dr. Dong-Gun Lee:

Thank you for the privilege of reviewing your work. Below you will find my comments, instructions from the Spectrum editorial office, and the reviewer comments which are included in the attached file. Please ensure these comments are fully addressed in a revision. The reviewer's point the lack of resolution in MLST to draw conclusions about identity must be addressed; MLST loci are a small subset of the genome, so conclusions on genetic identity can not be drawn from MLST analysis. While the MLST loci may be identical, it can not be concluded that the rest of the genome is also identical - this would require WGS. Also, the reviewer highlights that the use of the term clade is not appropriately used for the data here, the reviewers suggestion of clusters is more appropriate.

Revision Guidelines

Sincerely,
Christina Cuomo
Editor
Microbiology Spectrum

Reviewer #2 (Comments for the Author):

see uploaded document

I received the manuscript that has been renamed “Genetic relationships of *Aspergillus fumigatus* in hospital settings during COVID-19” for review. I appreciate that the authors made a significant effort in toning down the claims and updating some incorrect claims.

A few significant issues are still remaining however;

Line 38 Genetically identical or related

The currently used method does not have the resolution to observe whether two isolates are genetically identical.

Line 39, genotype refer to genes, but in this study, microsatellites are determined, or are the authors referring to the *cyp51A* genotype? It is not clear.

Line 140-141, results were consistent, can the authors be more specific? All three independent multiplex PCRs resulted in 100% identical values for all samples? If this is the case, why are 3 independent measurements needed, if not, then what is considered ‘consistent’?

Line 187, 490, genotype refer to genes, but in this study, microsatellites are determined, so they means a sequence type?

Line 201-204, there is no evidence for clades in the *A. fumigatus* group, so these need to be referred to as clusters!

Line 201, the clade -> clusters are numbered but I cannot see this numbering in the table of Figure 4. I can count from top to bottom, but it would be useful for the reader to be sure that this is the correct number of the clusters.

Line 205, what is considered a close temporal or spatial proximity? Can the authors specify this?

Figure 3, I assume that isolate F669 is also azole resistant with a TR46, but it does not have a black arrow (already has a red arrow). Maybe the authors can consider a large overlay on the whole circle itself (e.g. striped) to point out which isolates are azole susceptible. The arrows now indicate different types of data, red is patient isolates (I assume, but explanation of the red arrow is missing in the figure legend), which is already covered by the colour of the circles (so double) and the black arrows are rather small and indicate azole susceptibility.

Line 220, the authors claim that the results possibly indicate widespread nosocomial infections via contaminated air.

I have a few issues with this statement. First, what is considered contaminated air, since this fungus has a ubiquitous presence in our environment. Is that because the air in the hospital does not meet a minimum level of certain quality parameters? Is the spore load in the air higher than would be acceptable? Second, this is quite a bold statement since it goes against the dogma of pathogenesis of this fungus. Spores of this fungus are inhaled on a daily basis, so when admitted to the hospital, the local air of the

hospital is inhaled. Does that mean that this is now a hospital or nosocomial infection? Generally, infections with *A. fumigatus* are not considered hospital acquired. I would however be interested to discuss whether this is the case, but the data of this manuscript is not sufficient to debate or reconsider the current dogma's. Please remove this sentence.

Line 221, 258 as indicated above, the current method does not have the resolution to state that isolates are identical, please remove.

Responses to Editor's and Reviewers' Comments : Manuscript ID [Spectrum01902-24R1]

We would like to thank the editor and reviewers for the invaluable comments that helped a great deal in improving the overall quality of our original manuscript. Please find attached our revised paper and below our point-by-point responses to your insightful suggestions and valuable comments.

Reviewer: I received the manuscript that has been renamed "Genetic relationships of *Aspergillus fumigatus* in hospital settings during COVID-19" for review. I appreciate that the authors made a significant effort in toning down the claims and updating some incorrect claims.

A few significant issues are still remaining however;

Comment 1: Line 38 Genetically identical or related. The currently used method does not have the resolution to observe whether two isolates are genetically identical.

RESPONSE: Thank you for your insightful comments. As noted, the MLVA method does not provide sufficient resolution to determine genetic identity, as can be achieved using techniques like WGS. However, STR analysis is a highly reliable method that is widely used in applications such as paternity testing and phylogenetic classification. Accordingly, we have toned down this section and revised it to focus on relatedness. **Please see lines 37-38.**

Comment 2: Line 39, genotype refer to genes, but in this study, microsatellites are determined, or are the authors referring to the *cyp51A* genotype? It is not clear.

RESPONSE: Thank you for your valuable feedback. To ensure scientific accuracy and enhance reader comprehension, we revised the text as follows: **Line 39-40**, two CAPA clinical strains sharing MLVA sequence types and azole-resistant mutations were isolated in the same COVID-19 ICU four months apart.

Comment 3: Line 140-141, results were consistent, can the authors be more specific? All three independent multiplex PCRs resulted in 100% identical values for all samples? If this is the case, why are 3 independent measurements needed, if not, then what is considered 'consistent'?

RESPONSE: We performed multiple analyses to ensure the accuracy of the experimental results, and all three analyses consistently yielded the same results. By "same results," we mean that rather than analyzing the same multiplex PCR product three times, we conducted three independent PCRs (to verify PCR error) and confirmed that the number of tandem repeats (TR values) in the MLVA analysis was identical across all three. These three independent PCR analyses were performed as biological replicates to validate the reproducibility and reliability of the experiment [Nature Methods, 11(9), 879-880]. Since all results were identical, we presented them as a single outcome in the manuscript without additional mention.

Comment 4: Line 187, 490, genotype refer to genes, but in this study, microsatellites are determined, so they means a sequence type?

RESPONSE: Thank you for your valuable feedback. The term "genotype" could be interpreted as representing the entirety of genes in a strain, which does not align with the scope of this study, where classification is based on partial gene sequences. Therefore, we revised the term to "sequence type

(ST)." **Please see lines 186, 485, and 488.**

Comment 5: Line 201-204, there is no evidence for clades in the *A. fumigatus* group, so these need to be referred to as clusters!

RESPONSE: Thank you for your detailed review and thoughtful feedback. We revised the content as follows:

In Figure 3, "cluster" has been changed to "group," and in Figure 4, "clade" has been revised to "cluster."

All corresponding sections in the manuscript have been updated accordingly. The revisions are as follows:

Cluster → Group (Lines 184, 185, 193, and 194)

Clade → Cluster (Lines 200, 201, 203, 208, 209, 211, 213, 496, 498, and 500)

Comment 6: Line 201, the clade → clusters are numbered but I cannot see this numbering in the table of Figure 4. I can count from top to bottom, but it would be useful for the reader to be sure that this is the correct number of the clusters.

RESPONSE: Thank you for your valuable feedback. In accordance with your comment, we have replaced the term "clades" with "clusters" in Figure 4 and included the corresponding cluster numbers. Additionally, we have formatted the cluster column in Figure 4 with clear separations and annotated the cluster numbers at the top left of each section. We believe this revision will enable readers to more easily follow and understand the data. **Please see the revised Figure 4.**

Comment 7: Line 205, what is considered a close temporal or spatial proximity? Can the authors specify this?

RESPONSE: In Cluster 3, the MLVA revealed genetically similar strains that were repeatedly cultured at 1- to 2-month intervals within the same unit or adjacent spaces. Unlike the immediate culturing observed *in vitro* after contact, clinical infections may involve latent or opportunistic infections depending on the host's condition, leading to potential temporal associations. The repeated culturing of strains within the same unit or across floors further suggests possible spatial associations. By presenting the temporal and spatial information of each strain in Figure 4, we believe readers will find it easier to understand these patterns. **Please see the revised Figure 4.**

Comment 8: Figure 3, I assume that isolate F669 is also azole resistant with a TR46, but it does not have a black arrow (already has a red arrow). Maybe the authors can consider a large overlay on the whole circle itself (e.g. striped) to point out which isolates are azole susceptible. The arrows now indicate different types of data, red is patient isolates (I assume, but explanation of the red arrow is missing in the figure legend), which is already covered by the colour of the circles (so double) and the black arrows are rather small and indicate azole susceptibility.

RESPONSE: Thank you for your detailed review and valuable feedback. We agree that the markers and legend in Figure 3 could potentially confuse readers. To address this issue, we have revised Figure

3 and updated its legend accordingly. Additionally, we have included the relevant explanation in **Lines 485-492**.

“Circles are color-coded by source: red represents isolates from patients with COVID-19, light green represents isolates from patients without COVID-19, and blue represents environmental samples. Circle size indicates the number of isolates sharing the same sequence type. Lines connecting the circles depict genetic distances, with bold lines representing a 1-marker difference, solid lines a 2-marker difference, and dotted lines a 3-marker difference. Azole-resistant strains are marked by TR mutations, along with their corresponding strain numbers. TR, tandem repeat.”

Comment 9: Line 220, the authors claim that the results possibly indicate widespread nosocomial infections via contaminated air. I have a few issues with this statement. First, what is considered contaminated air, since this fungus has a ubiquitous presence in our environment. Is that because the air in the hospital does not meet a minimum level of certain quality parameters? Is the spore load in the air higher than would be acceptable? Second, this is quite a bold statement since it goes against the dogma of pathogenesis of this fungus. Spores of this fungus are inhaled on a daily basis, so when admitted to the hospital, the local air of the hospital is inhaled. Does that mean that this is now a hospital or nosocomial infection? Generally, infections with *A. fumigatus* are not considered hospital acquired. I would however be interested to discuss whether this is the case, but the data of this manuscript is not sufficient to debate or reconsider the current dogma's. Please remove this sentence.

RESPONSE: Thank you for your detailed review and thoughtful comments. We deeply agree with your comments that fungal spores are generally present in the air, making it challenging to distinguish between community-acquired and nosocomial infections. However, given the objectives of our study—to emphasize the possibility of nosocomial transmission and the importance of preventive measures based on prior and current findings—we have decided to retain the current description in the discussion for the following reasons:

1. Presence of fungal spores in hospital environments: While fungal spores can exist in environmental air, hospital environments are designed to control such contamination through HEPA filters and ventilation systems. Previous studies from our institution have also demonstrated the effectiveness of these environmental controls [J Clin Microbiol 57(7): e02023-02018.]. However, during the urgent response to the COVID-19 pandemic, sufficient environmental control and validation were lacking. In such scenarios, our study identified *Aspergillus* spores even within patient rooms and corridors, which we termed as "contaminated." Similar terminology has been used in other studies investigating *Aspergillus* spores in hospital environments during construction or contamination events [Clin Infect Dis 68(2): 321-329., BMJ Open 7(11): e018109.].

2. Community vs nosocomial infections: We acknowledge the conflicting results of prior studies on COVID-19-associated aspergillosis concerning community-acquired versus nosocomial origins [J Fungi (Basel) 9(3): 298.]. Nevertheless, our study leveraged MLVA to demonstrate genetic relatedness between *Aspergillus* strains isolated from patients with COVID-19, hospital environments, and other hospitalized patients. Similar studies have also defined nosocomial infections based on genetic similarities between clinical and environmental strains, and we believe this approach is both logical and appropriate [Clin Infect Dis 68(2): 321-329, Crit Care 24(1): 538.].

Given these reasons, all authors have carefully deliberated and unanimously decided to retain the current wording. We believe these findings and terminologies are significant, particularly in the context of preventing fungal colonization and infection in high-risk patients during a pandemic. We kindly request your understanding of this perspective.

Comment 10: Line 221, 258 as indicated above, the current method does not have the resolution to state that isolates are identical, please remove.

RESPONSE: Thank you for your thoughtful comments. As mentioned earlier, while this method does not achieve the high resolution of techniques such as WGS, it is scientifically valid for demonstrating genetic relatedness. Accordingly, we have toned down the statements and revised the text. **Please see lines 220 and 257.**

Ultimately, we would like to express our sincere gratitude to the editors and reviewers for their positive and constructive criticism. The manuscript has vastly benefited from your valuable and insightful comments and suggestions. We look forward to hearing from you and would be happy to address any further concerns, if required. We hope this further pushes the manuscript closer to publication in your esteemed journal.

Re: Spectrum01902-24R2 (Genetic relationships of *Aspergillus fumigatus* in hospital settings during COVID-19)

Dear Dr. Dong-Gun Lee:

Your manuscript has been accepted, and I am forwarding it to the ASM production staff for publication. Your paper will first be checked to make sure all elements meet the technical requirements. ASM staff will contact you if anything needs to be revised before copyediting and production can begin. Otherwise, you will be notified when your proofs are ready to be viewed.

Sincerely,
Christina Cuomo
Editor
Microbiology Spectrum